# Calcium dependence of both lobes of calmodulin is involved in binding to a cytoplasmic domain of SK channels

David B Halling*, Ashley E Philpo, Richard W Aldrich*

Department of Neuroscience, The University of Texas at Austin, Austin, United States

**Abstract** KCa2.1–3 $Ca^{2+}$-activated $K^+$-channels (SK) require calmodulin to gate in response to cellular $Ca^{2+}$. A model for SK gating proposes that the N-terminal domain (N-lobe) of calmodulin is required for activation, but an immobile C-terminal domain (C-lobe) has constitutive, $Ca^{2+}$-independent binding. Although structures support a domain-driven hypothesis of SK gate activation by calmodulin, only a partial understanding is possible without measuring both channel activity and protein binding. We measured SK2 (KCa2.2) activity using inside-out patch recordings. Currents from calmodulin-disrupted SK2 channels can be restored with exogenously applied calmodulin. We find that SK2 activity only approaches full activation with full-length calmodulin with both an N- and a C-lobe. We measured calmodulin binding to a C-terminal SK peptide (SKp) using both composition-gradient multi-angle light-scattering and tryptophan emission spectra. Isolated lobes bind to SKp with high affinity, but isolated lobes do not rescue SK2 activity. Consistent with earlier models, N-lobe binding to SKp is stronger in $Ca^{2+}$, and C-lobe-binding affinity is strong independent of $Ca^{2+}$. However, a native tryptophan in SKp is sensitive to $Ca^{2+}$ binding to both the N- and C-lobes of calmodulin at $Ca^{2+}$ concentrations that activate SK2, demonstrating that the C-lobe interaction with SKp changes with $Ca^{2+}$. Our peptide-binding data and electrophysiology show that SK gating models need deeper scrutiny. We suggest that the $Ca^{2+}$-dependent associations of both lobes of calmodulin to SKp are crucial events during gating. Additional investigations are necessary to complete a mechanistic gating model consistent with binding, physiology, and structure.

*For correspondence:
brentbert2000@gmail.com
(DBH);
raldrich@mail.utexas.edu (RWA)

## Editor's evaluation

This manuscript provides compelling evidence that in response to calcium, both lobes of the protein calmodulin change their interaction with a domain of a potassium channel. These findings provide valuable information about the molecular mechanics by which calcium binding is transduced to channels. Calmodulin is important for many biological processes and this work is expected to be of interest to researchers studying biophysics, protein conformational change, calcium signaling, and general physiology.

## Introduction

SK channels modulate cell membrane excitability and help to maintain cytosolic $Ca^{2+}$ homeostasis. Mutations in an SK gene can cause diseases including Zimmermann–Laband syndrome and idiopathic noncirrhotic portal hypertension (*Bauer et al., 2019*; *Koot et al., 2016*). Located on the plasma membrane, SKs open to release cellular $K^+$ in response to $Ca^{2+}$ influx to the cell after membrane depolarization. $K^+$ efflux helps to restore resting membrane potential, which can result in inactivation of $Ca^{2+}$ entry into the cell (*Barrett and Barret, 1976*; *Hugues et al., 1982*; *Moolenaar and Spector, 1979*). Thus, SKs operate in the $Ca^{2+}$ feedback cycle to fine tune intracellular $Ca^{2+}$ signals originating

**Figure 1.** CaM as an SK Ca²⁺ sensor. (**A**) Cartoon representation of the SK4 K⁺ ion channel shown as cylindrical helices and molecular surface depictions of CaM (pdb 6CNN). For clarity, only two of four subunits and CaMs are shown. Subunit 1 is yellow and subunit 3 is green. The N-lobe surface of CaM is vermillion, and the C-lobe surface of CaM is blue. Transmembrane helices are shown as S1–S6, and SK C-domain helices are shown as HA–HC. The peptide used in the binding studies, SKp, includes HA, HB, and part of HC. Diagonal subunits of SK4 bind a CaM in EM structure. Intracellular helix 4–5 A between S4 and S5 of transmembrane segments bind in a pocket of the N-lobe of CaM. C-domain helices HA and HB of the cross-subunit bind the C-lobe of CaM. (**B**) Bottom views of SK4 showing all four subunits showing CaMs bound, left side, or with CaM models removed, right side. The colors are the same as in panel A. The subunits that are not depicted in A are gray. On the right side, the SK sites that bind CaM are colored the same as the CaM lobes they contact. Panels (**C–F**) demonstrate the lobes of CaM can reorient to form diverse conformation states. The lobes of CaM can when bound to SK4 (pdb 6CNN) form a W-shape in C, when not bound to protein (pdb 1CLL) form an S-shape in D, when bound to an SK2 peptide (pdb 1G4Y) form a C-shape in E, or when bound to an MLCK peptide (pdb 1CDL) form an O-shape in F. (**G**) The N-sensor hypothesis of SK activation by CaM claims the N-lobe of CaM to be the Ca²⁺ sensor. The N-lobe binds a Ca²⁺-dependent site when Ca²⁺ binds to it. The C-lobe remains anchored to a site with no local change. In the NC-sensor hypothesis, both lobes of CaM are functional Ca²⁺ sensors. At trace Ca²⁺, the C-lobe dominates the interaction. At high Ca²⁺, a different binding mode is revealed that binds both Ca²⁺-bound lobes. The binding sites for each lobe of CaM need not be on the same subunit and the depiction is meant to emphasize that SK responds to Ca²⁺ binding to both lobes of CaM.

at the plasma membrane. Yet $Ca^{2+}$ does not interact physically with SK. $Ca^{2+}$ activates the channel via another protein modulator that tethers itself to diagonal subunits of a homotetrameric SK channel (*Figure 1A, B*).

Calmodulin (CaM) directs the $Ca^{2+}$ response in virtually all eukaryotic cells. It binds hundreds of targets that include ion channels. Different properties from each of its four unique $Ca^{2+}$-binding sites are highly conserved (*Halling et al., 2016*) leading to dynamic ranges of $Ca^{2+}$ sensitivity that can vary depending on which target CaM is bound to *Black et al., 2005*; *Newman et al., 2008*; *Persechini et al., 2000*; *Westerlund and Delemotte, 2018*. The four $Ca^{2+}$ sites are arranged into two pairs, each pair comprising a domain or 'lobe' of CaM. The two globular lobes form a 'dumbbell' like appearance (*Babu et al., 1985*; *Figure 1D*). The two $Ca^{2+}$ sites within each lobe may tune each lobe to respond differently to $Ca^{2+}$ stimuli (*Liang et al., 2003*).

Each CaM target has evolved to use different properties of a versatile CaM architecture (*Figure 1C–F*). A common challenge encountered with mechanistic studies on CaM's targets is knowing to what extent the lobes of CaM sense $Ca^{2+}$ independently (*Banerjee et al., 2018*; *Schumacher et al., 2001*; *Tadross et al., 2008*; *Williams et al., 2018*) or together as intercommunicating units to carry out a function (*Evans et al., 2011*; *Mukherjea et al., 1996*; *Persechini et al., 1996*). Changes in cytosolic $Ca^{2+}$ alter how a tethered CaM binds to SK (*Xia et al., 1998*). For SK gating, a simple model that was based on structures, electrophysiology, and some binding data suggest that the C-terminal domain (C-lobe) of CaM is constitutively bound to SK independent of $Ca^{2+}$ (*Keen et al., 1999*; *Lee and MacKinnon, 2018*; *Schumacher et al., 2001*). The N-terminal domain (N-lobe) forms the $Ca^{2+}$ sensor of SK and only binds tightly in the presence of $Ca^{2+}$. We refer to domain-dependent mechanism as the 'N-sensor hypothesis' for SK gating (*Figure 1G*). Although the N-sensor hypothesis has an experimental basis, it has not been thoroughly tested.

Certain properties must hold to validate the N-sensor hypothesis for SK: (1) The C-lobe is insensitive to $Ca^{2+}$ when bound to SK. (2) The N- and C-lobes act independently and carry out distinct functions, that is, the N-lobe is the $Ca^{2+}$ sensor, whereas the C-lobe is the $Ca^{2+}$-independent glue to keep CaM bound to SK. (3) The N- and C-lobes bind at different sites so that their unique functions can be carried out. In contrast, many CaM-binding domains bind both lobes in proximity and interact with other either directly or through peptide contacts (*Meador et al., 1992*; *Villarroel et al., 2014*). Recent structures of SK4 with bound CaM show multiple conformations of the N-lobe at trace $Ca^{2+}$, but a better resolved interaction at high $Ca^{2+}$ (*Lee and MacKinnon, 2018*). At high $Ca^{2+}$ the N- and C-lobes of CaM bind to different domains of SK4 in a unique conformation for CaM that has not been observed previously (*Figure 1A, B*). Given the diversity of interactions in the structures, careful binding studies will be needed to sort out how each domain of CaM interacts with SK at low and at high $Ca^{2+}$.

Although CaM modulation of SK has been studied for decades, there are gaps in our fundamental understanding that still need to be worked out. Transduction of $Ca^{2+}$ binding to channel gating is a dynamic, multistep process. It includes $Ca^{2+}$ binding to CaM, CaM binding to SK, and the intrinsic motions of SK through different gating states.

## SK function can be studied by exchanging WT-CaM with mutant CaM

WT-CaM has a high affinity for SK2 at both high and trace $Ca^{2+}$ (*Xia et al., 1998*). In a heterologous system, WT-CaM is always present, so we developed a method that allowed us to exchange WT-CaM for a mutant CaM (*Li et al., 2009*; *Figure 2A*). At 10 µM $Ca^{2+}$, SK2 approaches maximal open probability in the presence of WT-CaM. To study how different CaM constructs can restore SK2 currents, we first coexpressed the SK channel with a mutant CaM (E1Q) that has a glutamate instead of a glutamine at the conserved position 12 in the N-lobe EF hand 1, which is residue 31 in the protein sequence. This mutation leads to only one functional EF hand in the N-lobe (*Maune et al., 1992*). We demonstrated (*Li et al., 2009*) that coexpression of E1Q with SK2 in *Xenopus laevis* oocytes leads to a rapid rundown of SK2 current. Current rundown can be recovered permanently upon application of exogenous WT-CaM, that is SK2 current no longer runs down once WT-CaM associates with the channel. We assume that exogenous CaM binds to empty CaM sites on SK from E1Q dissociation, or alternatively E1Q leaves the C-lobe bound and that exogenous CaM can displace the mutant. These findings gave us a tool to test whether SK channel activity could be recovered by various non-WT-CaMs.

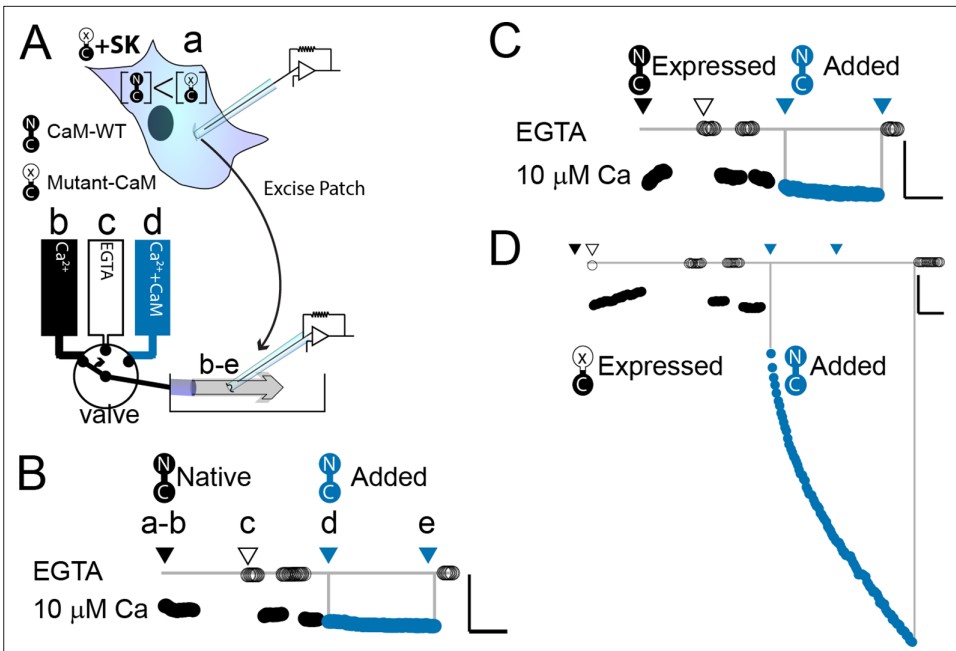

**Figure 2.** WT-CaM recovers SK2 current in HEK293 cells. (**A**) Cartoon depicts patch excision at time zero (**a**) and moving an electrode into the flow path from a fast solution exchanger (**b–d**). For simplicity, only three of eight solution positions are depicted. The illustration in (**A**) best describes the conditions for the experiment in panel (**D**). The scale bars in B–D indicate 0.13 nA and 50 s for horizontal and vertical lines, respectively. Average current level at −60 mV measured every 3 s from a patch excised from HEK 293 cells transfected with rSK2 only in (**B**), cotransfected with excess WT-CaM in (**C**), or cotransfected with excess mutant CaM (E1Q) in (**D**). Note that in (**B**) rSK2 contacted endogenous CaM, and that in both (**C**) and (**D**) WT- and mutant CaM are massively overexpressed relative to endogenous CaM. Traces show rSK2 current before (black) and after (blue) application of exogenous WT-CaM protein. Time point labels in (**B–D**) are the same as in (**A**): a. patch excision, b. movement of electrode to the Ca$^{2+}$ solution-exchange stream, c. first application of EGTA (Ethylene glycol-bis(β-aminoethyl ether)-N,N,N′,N′-tetraacetic acid tetrasodium salt) to find basal patch current, d. timepoint at which purified, recombinant WT-CaM applied shown in blue trace, e. timepoint at which fractional recovery is measured. WT-CaM is indicated by a dumbbell with an N- and a C-lobe. E1Q is depicted as a dumbbell with an 'x' in the upper lobe to indicate where Ca$^{2+}$ binding is disrupted.

## Binding studies provide information on a Ca$^{2+}$ dependence of interactions

SK2 gating requires CaM to bind its C-terminus (*Schumacher et al., 2001*; *Xia et al., 1998*). Solution binding studies can be performed on protein domains of SK that bind CaM. A peptide of SK2 that comprises the CaM-binding domain is referred to as SKp (residues 396–487 of rat KCa2.2) has been used in multiple binding and structure studies (*Halling et al., 2014*; *Nam et al., 2017*; *Schumacher et al., 2001*; *Zhang et al., 2012a*; *Zhang et al., 2012b*; *Zhang et al., 2013*). We used multi-angle light scattering to show that full-length WT-CaM has unique properties with SKp that isolated, separated, or duplicated lobes of CaM cannot reproduce.

Intrinsic fluorescence from a peptide, such as SKp, can be used to analyze CaM binding (*Keen et al., 1999*). Tryptophan (W) has the strongest fluorescent properties of all the amino acids (*Lakowicz, 2006*). Its fluorescence is also sensitive to its environment. Although tryptophan emission spectra are subject to several factors, the comparisons of gross changes in its amplitude and the location of its peak intensity can be a reporter for molecular binding. In general, a spectrum with a peak at longer wavelengths suggests a solvent exposed tryptophan, whereas a peak at shorter wavelengths describes a tryptophan in a more polar environment, such as being buried in a protein. We used SKp-W432 spectra to show that either lobe of CaM can bury this residue.

We use both electrophysiology and peptide-binding data to show some properties of CaM that are necessary for CaM to bind SKp, and how these findings relate to CaM-dependent SK function.

**Table 1.** Quantification of effects exogenous CaM protein has on SK current recovery.

| CaM construct | n | Fraction of recovered current (median) | Fraction of recovered current (mean) | Standard deviation |
|---|---|---|---|---|
| WT | 25 | .95 | .86 | .21 |
| CaM (E34Q) | 4 | .54 | .50 | .16 |
| N-CaM | 4 | .02 | .05 | .09 |
| C-CaM | 4 | .007 | .03 | .08 |
| N-CaM + C-CaM | 5 | .05 | .07 | .06 |
| Double N-lobe | 4 | .09 | .08 | .04 |
| Double C-lobe | 4 | .21 | .19 | .12 |

## Results

### WT-CaM can recover SK channel activity in HEK293 cells

We wanted to explore what components of mammalian wild-type calmodulin (WT-CaM) are necessary for SK channel activation. In control experiments, patches excised from cells that were transfected with either SK2 alone or both SK2 and WT-CaM had minimal increase in SK current in 10 µM $Ca^{2+}$ when exogenous WT-CaM was applied for 297 ± 6 or 278 ± 41 s, respectively (SK alone, 21 ± 14%; $n = 4$; SK + WT-CaM 32 ± 13%; $n = 3$) (*Figure 2B, C*). Since adding exogenous CaM does not substantially increase current, all available SK channels in the excised patch are likely fully saturated with WT-CaM.

We conducted electrophysiological experiments to determine if non-wild-type CaMs could support SK channel activation at saturating (10 µM) calcium. Because our expression system currently is in HEK293 cells and the results from *Li et al., 2009* were produced from oocytes, we first validated the same approach in HEK293 cells. As expected, patches excised from cells that were cotransfected with SK and E1Q had an average of a ~sixfold increase in current level when exogenous 20 µM WT-CaM was applied for 300 s ($n = 3$) (*Table 1*, *Figure 2D*).

Like oocytes, HEK293 cells produce endogenous CaM that transfected E1Q protein must outcompete to produce the observed current rundown. If SK2 binds E1Q with lower affinity than WT-CaM, one explanation is that the exogenously applied WT-CaM binds to sites that were vacated by E1Q when the patch was excised. In the absence of E1Q, endogenous WT-CaM is sufficient to form functional units with SK2.

Although we did not quantify the extent of SK channel run down after patch excision, like what was done in oocytes, we are confident that as in oocytes the increase in SK current with the application of exogenous WT-CaM was due to the recovery of SK channel activity that had been lost by the overexpression of E1Q.

### SKp binds WT-CaM with different conformations at low and at saturating $Ca^{2+}$

The C-terminus of SK is required for CaM binding. We use a peptide to represent this site (SKp). Complex formation between SKp and CaM constructs in solution was measured using composition-gradient multi-angle light scattering (CG-MALS). CG-MALS measures the weight average molar mass of particles in solution as a function of molar ratio of the molecules of interest (*Attri and Minton, 2005*). Light-scattering signals are directly converted into a weight average molar mass if protein concentrations and instrument parameters are precisely known (*Wyatt, 1993*), thus quantifying higher-order stoichiometries is a designed capability of this approach. Our prior work shows that CaM and SKp form different stoichiometries that depend on the molar ratio concentrations of CaM to SKp in aqueous solution (*Halling et al., 2014*). In a 'cross-over' experiment, the sample applied to the flow cell in the light chamber varies using a series of injections to step from high molar concentrations of SKp relative to CaM crossing over to high molar ratios of CaM relative to SKp. Complex formation is measured at each step of varying protein ratios, thus capturing a wide range of possible stoichiometries in a single experiment. For simplicity, we refer to the stoichiometries as follows: (1) When the molar concentration of SKp exceeds WT-CaM, a complex with 2 SKp and 1 CaM (2SKp/1CaM) forms,

which we call 'P-C-P'. (2) At equimolar concentrations of SKp and WT-CaM it forms a 1SKp/1CaM complex, which we call 'P-C'. (3) When WT-CaM molar concentrations exceed SKp, 1 SKp and 2 WT-CaMs (1SKp/2CaM) form a complex we call 'C-P-C'.

SKp has a molar mass of 11.1 kDa. We repeated SKp-binding measurements with WT-CaM at low and at high $Ca^{2+}$ as a control to draw comparisons with different CaM constructs (*Figure 3A*). WT-CaM has a molar mass of 16.7 kDa. At 5 mM $Ca^{2+}$, the characteristic appearance of the calculated weight-averaged molar masses shows an 'M' shape as a function of the molar ratio of WT-CaM protein to SKp. The 'M' shape is fortuitous in our studies because the features, that is, peaks and valleys, constrain models that determine what stoichiometries WT-CaM and SKp can form in solution. The light-scattering data are decomposed into molar fractions of free and bound protein that are present in solution giving rise to the features. The data confirm that CaM forms three different complexes (*Figure 3K*), P-C-P, C-P, and C-P-C.

Fitting the data to a model provides ranges on macroscopic association constants. Since these data are weight averages, the C-P-C peak is smaller because it forms with weaker affinity than the P-C-P complex, with a log $K_A$ of 16.0 and 17.0 (p < 0.01), respectively (*Table 2*). We note one difference from our previous results (*Halling et al., 2014*), fitting for incompetent fraction did not provide substantial improvements to our fits, so our data this time were fit with simpler models and with fewer parameters. Eliminating the incompetent fraction allowed us to use the minimum number of variables to avoid parameter compensation that could lead to non-unique solutions for association constants.

Metal chelators, such as EDTA (Ethylenediaminetetraacetic acid) and EGTA are commonly used to reduce free metal in solution, but the quantities of free metal ions in solutions are usually calculated or assumed. When we use EGTA, we consider it a 'trace $Ca^{2+}$' condition with $Ca^{2+}$ concentrations assumed to be nanomolar, but metal contaminants are unknown. At trace $Ca^{2+}$, there are fewer distinguishing features (*Figure 3A, F*). Notably P-C-P is at much smaller quantities, and C-P-C is absent as observed previously (*Halling et al., 2014*). Just as we observed with $Ca^{2+}$, in EGTA P-C has a high affinity. Since the P-C complex has negligibly different masses at high and at trace $Ca^{2+}$, another approach is needed to observe whether $Ca^{2+}$ causes a change to the interaction of the peptide and CaM.

## Intrinsic tryptophan fluorescence of SKp confirms calcium sensitivity of calmodulin

In the absence of a binding partner, the SKp W432 emission spectrum has a peak position at or near 348 nm in all buffers used (*Figure 4A–C*). The appearance of this emission spectrum is consistent with earlier reports, and it suggests that in the absence of CaM the tryptophan in SKp is solvent exposed (*Keen et al., 1999*).

In the presence of WT-CaM equimolar with SKp, the emission peak from W432 of SKp shifts to shorter wavelengths (*Figure 4A–C*). At trace $Ca^{2+}$, with 5 mM EGTA, the W432 peak is at 334 nm, and it has a 230% greater amplitude than SKp alone (*Figure 4A*).

In our electrophysiology experiments, SK2 has maximal opening at 10 µM free $Ca^{2+}$. On the other hand, structures are usually solved at millimolar concentrations of $Ca^{2+}$. To draw comparisons to both electrophysiology and structure data, we measured $Ca^{2+}$-bound CaM binding to SKp at both 10 µM $Ca^{2+}$ (*Figure 4B*) and 5 mM $Ca^{2+}$ (*Figure 4C*). The spectrum of W432 at either $Ca^{2+}$ concentration is virtually the same. Compared to trace $Ca^{2+}$ the amplitude appears 30% weaker, and the peak is shifted toward shorter wavelengths (329 nm). We suspect a 310-nm cut-on emission filter contributes to the decrease in amplitude as the peak shifts toward smaller wavelengths, so the decrease in amplitude could be an artifact of our optics. Still, the pronounced change in spectral properties indicates both that W432 senses protein binding and that protein binding is sensitive to $Ca^{2+}$. These major differences in spectral shapes strongly suggest that the P-C complex has different conformations at low and at high $Ca^{2+}$.

We can use these features as we study other CaM constructs to determine what properties of CaM are required for SK2 activity and for $Ca^{2+}$-dependent interactions with SKp.

## SK requires CaM with both an N- and a C-lobe connected for activity to be recovered

Given that we could recover SK channel activity with the application of exogenous WT-CaM protein in HEK293 cells, we then tested whether the non-WT-CaM constructs could also recover SK activity.

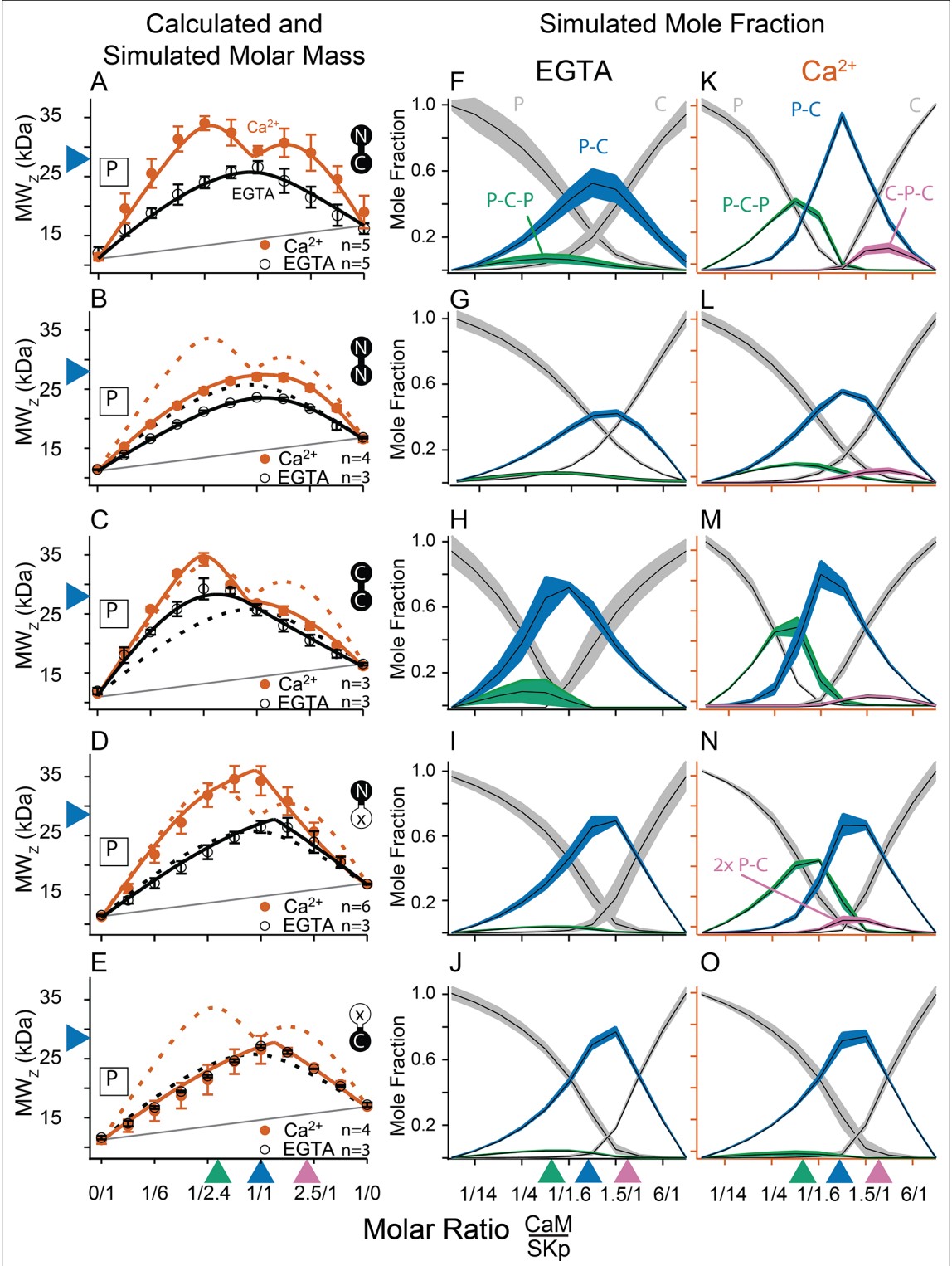

**Figure 3.** Composition-gradient multi-angle light scattering (CG-MALS) assays of SKp-binding full-length CaM constructs. A 1/1 complex of SKp/WT-CaM is 28 kDa and indicated by blue arrows on the left panels. Each row assays the complex formation of SKp (P) with a specific CaM construct indicated by labels: Row (**A**) WT-CaM, (**B**) Double N-lobe CaM, (**C**) Double C-lobe CaM, (**D**) C-lobe $Ca^{2+}$ mutant CaM, and (**E**) N-lobe $Ca^{2+}$ mutant CaM. (**A–E**) Weight average molar mass as a function of molar ratio of CaM:SKp, black circles 5 mM EGTA, vermillion circles 2 mM $CaCl_2$. Error bars show standard error of the mean. Solid curves connecting data points are simulations using association constants derived from fits. Gray lines show what points would follow if there was no interaction. Dashed lines in B–E are the same curves as in panel A for comparison. Panels (**F–J**) in 5 mM EGTA

*Figure 3 continued on next page*

*Figure 3 continued*

and (**K–O**) in 2 mM Ca$^{2+}$ show fitted mole fractions of different complexes required in the experimental solution to produce the data in (**A–E**). Shaded regions are within the standard error of the mean. The gray regions (P) and (C) correspond to fraction of unbound SKp (P) or CaM (C) protein. The arrows along the bottom indicate where the total molar ratios of SKp:CaM are 2:1 green, 1:1 blue, and 1:2 purple.

For each experiment we excised inside-out patches from cells that were cotransfected with SK and E1Q. We took each patch and calculated what fraction of the total recovered SK current was due to the application of our various mutant CaM constructs (*Figure 5A*). The constructs included an N-lobe only, a C-lobe only, a mixture of N- and C-lobes, a double N-lobe, a double C-lobe, and CaM with Ca$^{2+}$ binding disrupted in the C-lobe CaM (E34Q). The CaM (E34Q) mutant contains glutamate to glutamine (E to Q) substitutions at positions 104 and 140 (sites 3 and 4) in the C-lobe, CaM (E34).

We found that none of the non-WT-CaMs, except for the mutant CaM (E34Q), lead to a magnitude of recovery comparable to WT-CaM (*Figure 5B, C*, *Table 1*).

## SKp binds N- and C-lobe with high affinity

The absence of SK current recovery by a non-WT-CaM construct could indicate either that CaM can still bind SK but requires its full protein sequence to activate the channel, or that CaM binding is disrupted and the absence of CaM renders SK Ca$^{2+}$ insensitive. Binding measurements are necessary to determine whether our mutant CaM constructs can still bind to SK.

N-CaM has a molar mass of 9.0 kDa. Combined with SKp, which has a molar mass of 11.1 kDa, the molar mass of a 1/1 complex would be about 20 kDa. We used CG-MALS to determine if this stoichiometry, or other combinations, can form at high and at trace Ca$^{2+}$. Unlike the features seen with WT-CaM at high Ca$^{2+}$, N-CaM binding to SKp has a simple, triangular shape in CG-MALS data (*Figure 6A*). The apex of triangular feature approaches the predicted molar mass of a 1/1 complex. There are no bulges at other molar ratios that would indicate other stoichiometries can form. The triangular shape with the apex at the predicted molar mass shows a strong $K_A$ for a 1/1 complex and that the P-C complex is dominant in the solution (*Figure 6A, E*, *Table 2*). Similar extrapolated $K_A$ values between N-CaM and WT-CaM binding to SKp might indicate similar binding properties, but the interpretation could be misleading without additional work. WT-CaM has four Ca$^{2+}$ sites, but N-CaM only has two. Careful Ca$^{2+}$-binding measurements are needed to compare the $K_A$ values of mutant CaMs with different numbers of Ca$^{2+}$ sites to WT-CaM in binding to SKp. The key comparisons of the MALS data at high Ca$^{2+}$ are that WT-CaM has more ways to bind to SKp than N-CaM, and that both CaMs can bind SKp with strong affinity. In contrast at trace Ca$^{2+}$, N-CaM binding to SKp is noticeably weaker (lower $K_A$) and mostly unbound (*Figure 6A, C*, *Table 2*).

C-CaM forms the same stoichiometry regardless of Ca$^{2+}$ (*Figure 6B, D, F*). The association of C-CaM with SKp is strong, it is also independent of Ca$^{2+}$. N-CaM forms only a 1/1 complex with SKp

**Table 2.** Fitted association constants, $K_A$, from composition-gradient multi-angle light scattering (CG-MALS), shown as 'mean log $K_A$ (standard error) $n$', where $n$ = number of trials.

| SKp + CaM construct | 1SKp/1CaM (P-C) | | 2SKp/1CaM (P-C-P) | | 1SKp/2CaM (C-P-C) | | 2SKp/2CaM (2x P-C) | |
|---|---|---|---|---|---|---|---|---|
| | −Ca$^{2+}$ | +Ca$^{2+}$ | −Ca$^{2+}$ | +Ca$^{2+}$ | −Ca$^{2+}$ | +Ca$^{2+}$ | −Ca$^{2+}$ | +Ca$^{2+}$ |
| WT-CaM | 6.9 (0.2)5 | 10.4 (0.2)5 | 12.2 (0.1)5 | 17.0 (0.2)5 | – | 16.0 (0.3)5 | – | – |
| NN-CaM | *6.4 (0.1)3 | 7.0 (0.1)4 | *11.5 (0.1)3* | 12.5 (0.1)4 | – | 12.3 (0.1)4 | – | – |
| CC-CaM | 9.9 (0.1)3 | 9.7 (0.9)3 | 15.1 (0.5)3 | 16.5 (1.0)3 | – | 14.8 (0.9)3 | – | – |
| N-CaM | 5.1 (0.1)3 | *8.9 (0.7)3* | – | – | – | – | – | – |
| C-CaM | 7.6 (0.2)3 | *8.1 (0.7)3* | – | – | – | – | – | – |
| CaM (E12Q) | *8.2 (0.7)3* | *8.8 (0.8)3* | 13.2 (0.7)3 | 14.0 (0.9)3 | – | – | – | – |
| CaM (E34Q) | *8.3 (0.8)3* | *9.1 (0.1)6* | 13.1 (0.9)3 | 15.8 (0.1)6 | – | – | – | 23.3 (0.1)6 |

*Italicized values fall outside the limit of detection for modeling KA, or fits are ill constrained.

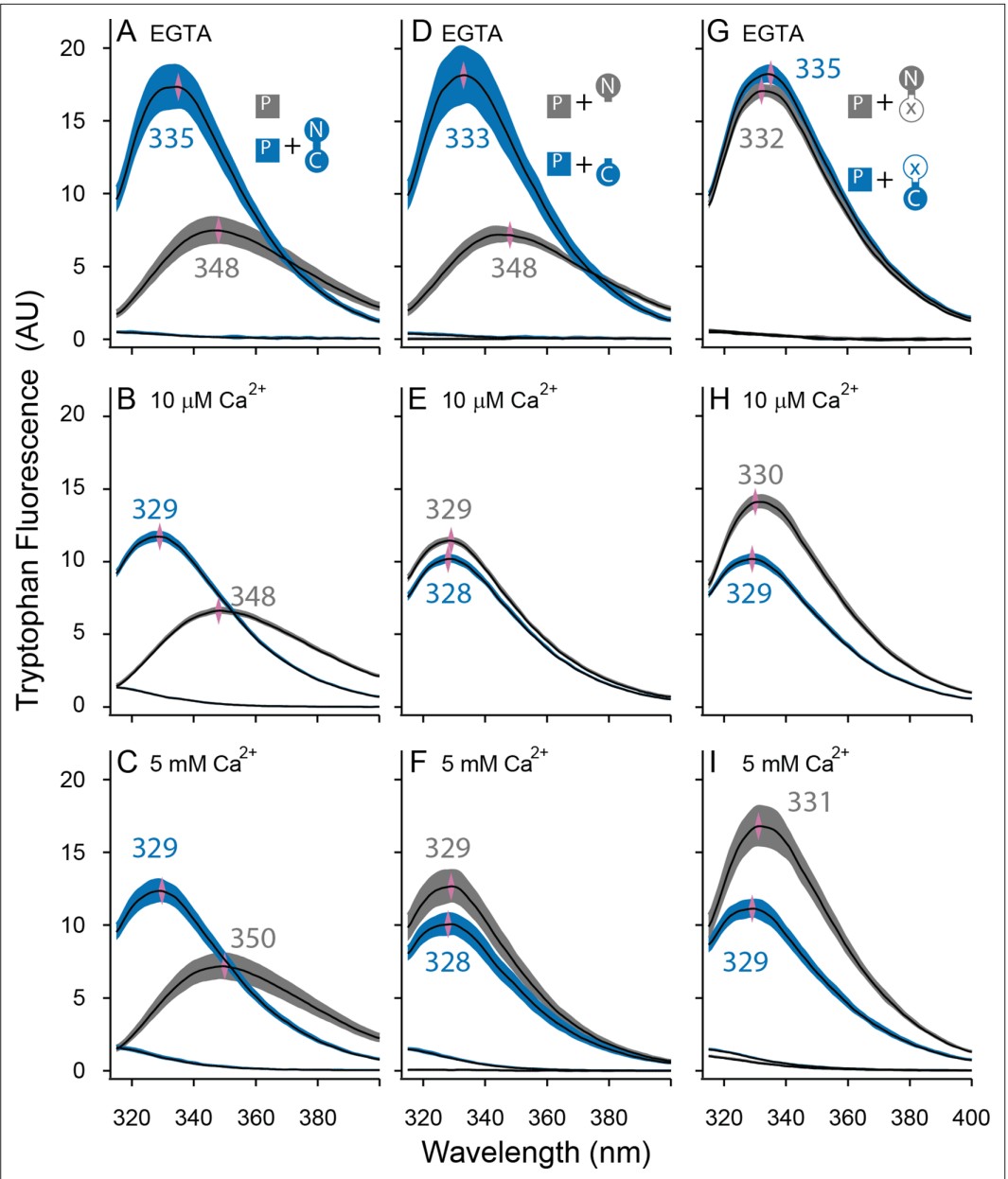

**Figure 4.** Both lobes can alter the environment of W432 of SKp. Shown are averaged fluorescent emission spectra from excitation at 265 nm. Shaded regions are within the standard error of the mean. Rows of panels have the same concentration of $Ca^{2+}$. Top row was measured in 5 mM EGTA (<5 nM $Ca^{2+}$), middle row at 10 μM free $Ca^{2+}$, bottom row in 5 mM $Ca^{2+}$. (**A–C**) SKp (P) emission spectra are shaded gray. SKp emission spectra with equimolar CaM are shaded blue. (**D–F**) SKp emission spectra with equimolar N-CaM (gray) or C-CaM (blue). (**G–I**) SKp emission spectra with equimolar CaM (E12Q) (gray) or CaM (E34Q) (blue). Controls of CaM emission without SKp present are included in most panels. Purple diamonds and nearby values indicate the wavelength (nm) where the peak average intensity was located.

when $Ca^{2+}$ is present. Additional approaches are needed to determine whether $Ca^{2+}$ alters the configuration of P-C for SKp binding to C-CaM.

## Both lobes of CaM have $Ca^{2+}$-dependent associations with SKp

We showed in *Figure 4A, B* that $Ca^{2+}$ alters the interaction of SKp with full-length CaM. We wanted to use this observation to assay whether $Ca^{2+}$ changes the configuration of N-CaM or C-CaM. We compared the spectra of SKp + WT-CaM with spectra of SKp + either N-CaM or C-CaM. At trace $Ca^{2+}$,

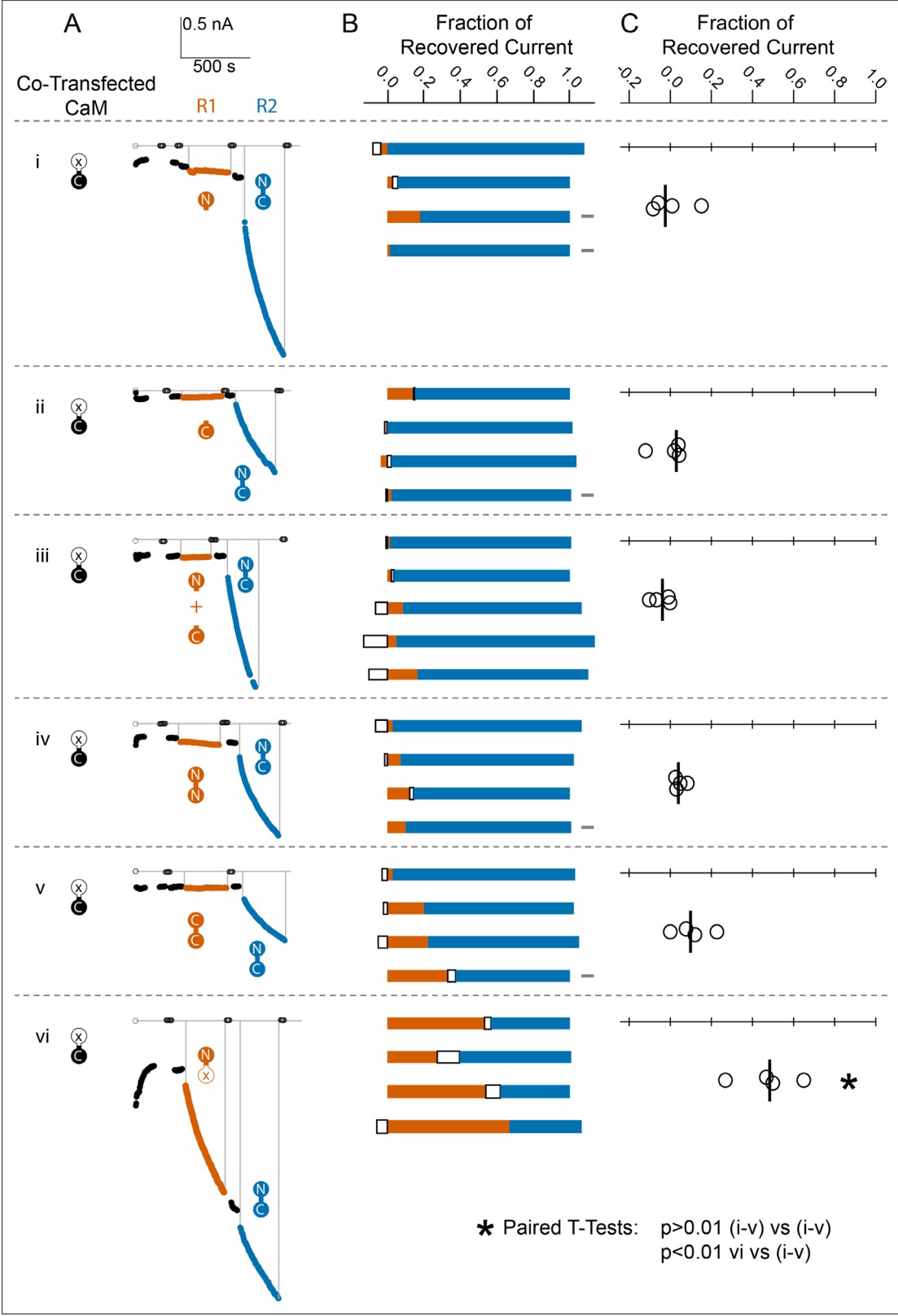

**Figure 5.** Comparison of fractional SK2 current recovery with various CaM constructs. Column (**A**): Average current levels at −60 mV measured every 3 s from patches excised from cells cotransfected (CT) with rSK2 and mutant CaM (E1Q). Basal EGTA (open black circles) and $Ca^{2+}$ (filled black circles) currents are established, exogenous non-WT-CaM protein is applied (R1 – vermillion traces), washed out (black traces again), WT-CaM protein is applied (R2 – blue traces), then washed out (black traces). Each row (**i–vi**) is an experiment with the non-WT-CaM as indicated by the vermillion labels under R1.

*Figure 5 continued on next page*

*Figure 5 continued*

Column (**B**): Each bar represents a different patch. The blue portion of the bar represents the fraction of current recovered by exogenous WT-CaM, the vermillion portion represents the fraction recovered by exogenous non-WT-CaM, and the unfilled portion represents the fraction of drift in current at 10 μM free $Ca^{2+}$ after application of non-WT-CaM and before application of WT-CaM. The gray rectangles to the right of certain experiments indicate that they deviated slightly from the experimental protocol described in the Materials and methods, primarily in solution changes before and after non-WT-CaM. Column (**C**): Fraction of SK current recovered with the application of non-WT-CaM constructs relative to WT-CaM. Each open black circle represents a different patch, and the black vertical bar is the median fractional recovery. In B, fractional recovery extended beyond 1 and below 0 in some experiments due to current drifting up or down slightly across solution changes and over the course of recovery. Similarly, in C, fractional recovery for some experiments was below 0 due to current decreasing slightly during application of non-WT-CaM construct. In paired *T*-tests, rows i–v were not significantly different from each other (each p > 0.01), but row vi was different from each other row (each p < 0.01).

the W432 spectra obtained with SKp bound to C-CaM is indistinguishable from when SKp is bound to WT-CaM (*Figure 4D*). In contrast, when N-CaM is present, the W432 spectrum is indistinguishable from when SKp is by itself, which is consistent with N-CaM's low affinity for SKp at trace $Ca^{2+}$. Thus, we conclude that the environment of W432 appears to be dominated by the C-lobe of WT-CaM at trace $Ca^{2+}$.

10 μM $Ca^{2+}$ is sufficient to fully activate SK2 channels (*Li et al., 2009*). Binding experiments at 10 μM $Ca^{2+}$ show that SKp bound to either isolated N-CaM or isolated C-CaM produces spectra that are very close to SKp + WT-CaM (*Figure 4B, E*), but there is a subtle, 10–20% smaller amplitude when the C-lobe is bound. The spectrum with SKp bound to N-CaM appears slightly more like the spectrum of SKp bound to WT-CaM, but not enough to conclude that it is the N-lobe of CaM that binds W432 when full-length WT-CaM binds SKp. For binding measurements, we wanted to make sure 10 μM $Ca^{2+}$ is saturating. 5 mM $Ca^{2+}$ does not change the spectra further (*Figure 4C, F*), indicating that 10 μM $Ca^{2+}$ is sufficient for maximal effect of $Ca^{2+}$ with either or both lobes.

## C-P-C only forms with full-length CaM

At high $Ca^{2+}$ SKp forms the C-P-C complex with WT-CaM (*Figure 3A, F*). The C-P-C complex is not observed with either N-CaM or C-CaM using the cross-over approach (*Figure 6*). We wanted to make sure that a C-P-C complex with N- or C-CaM was not missed due to weaker binding. To

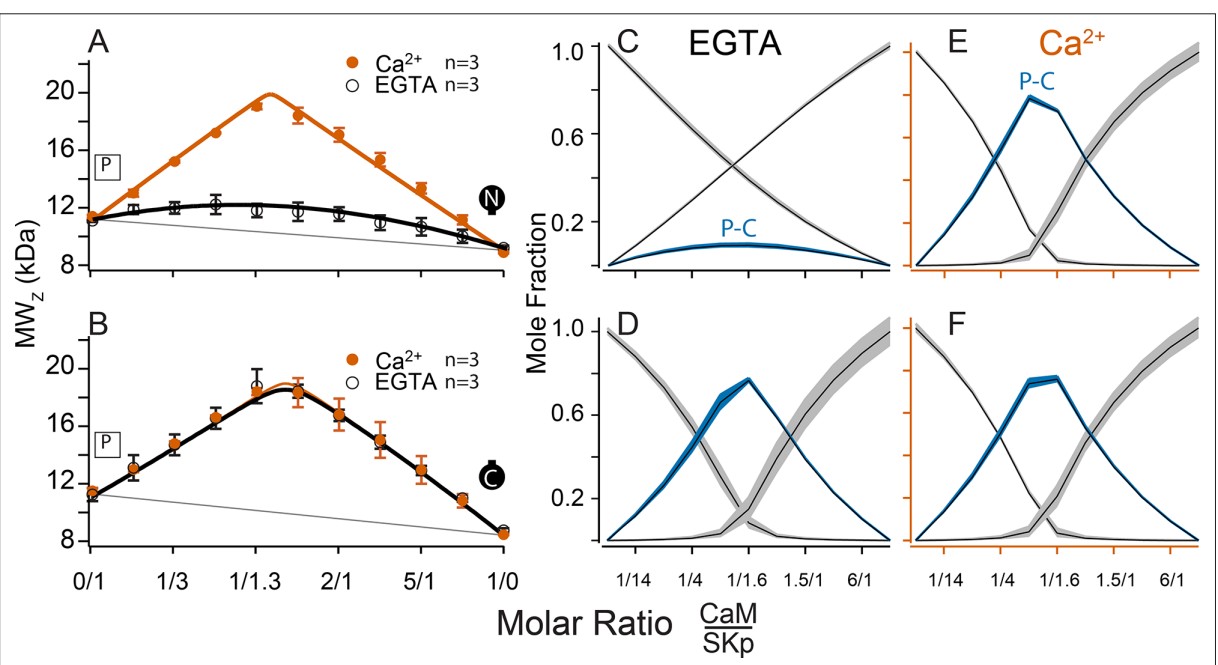

**Figure 6.** Composition-gradient multi-angle light scattering (CG-MALS) assays of SKp binding to half-CaM molecules. (**A, B**) Weight average molar mass as a function of molar ratio of half-CaM:SKp, black circles 5 mM EGTA, vermillion circles 2 mM $CaCl_2$. Error bars show standard error of the mean. Lines connecting data points are simulations using association constants calculated from fits. Panels (**C, D**) in 5 mM EGTA and (**E, F**) in 2 mM $Ca^{2+}$ show fitted mole fractions of free or bound proteins using the data in A, B. Shaded regions around the curve are within the standard error of the mean.

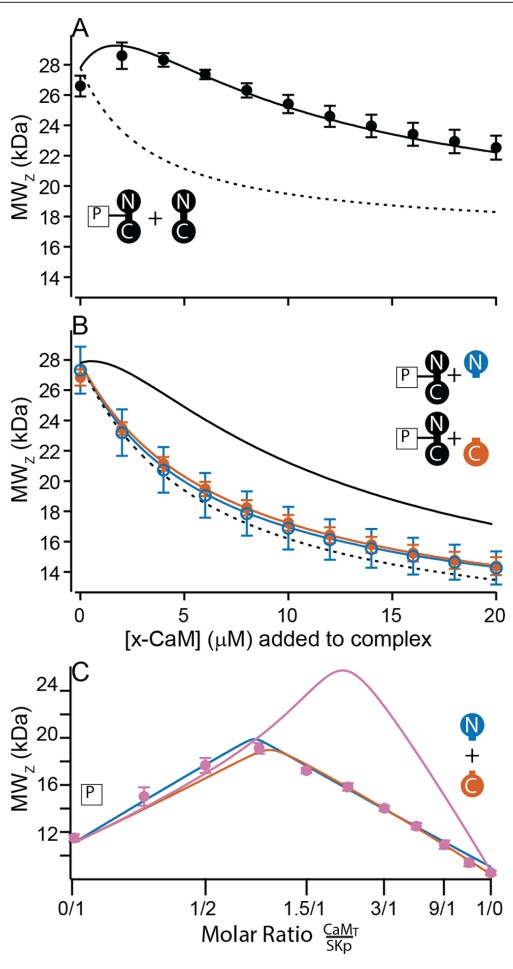

**Figure 7.** Only full-length CaM can bind to form the C-P-C complex. (**A, B**) Weight average molar mass of complex as a function of added CaM. Note that the weight average molar mass decreases at higher concentrations of unbound protein relative to a complex. Full-length, WT-CaM was premixed with SKp at a 1:1 molar ratio to form the P-C complex. The same amount of P-C complex is mixed with increasing concentrations of WT-CaM (**A**) or half-CaM (**B**) N-CaM open blue circles, C-CaM closed vermillion circles. Solid line shows predicted weight average molar mass of C-P-C at higher concentrations of CaM using association constants used from fits to *Figure 3A*. The dashed line shows prediction if P-C stays bound but no additional molecules bind. (**C**) Weight average molar mass, closed circles, as a function of molar ratio of total CaM:SKp at 2 mM $Ca^{2+}$. $CaM_T$ (total CaM) has equal parts of N-CaM and C-CaM. Blue line is the simulation N-CaM binding to SKp from *Figure 6A*, vermillion line C-CaM binding to SKp from *Figure 6B*, and purple line is a simulation if both N-CaM and C-CaM bind SKp together.

sample data at high molar ratios of CaM relative to SKp, we designed experiments that would emphasize a $Ca^{2+}$-dependent site. We used the high-affinity P-C complex of SKp and WT-CaM as a tool. Beginning with the preformed P-C and holding the P-C concentration at a constant value, we titrated increasing concentrations of WT-CaM. At low additions of WT-CaM there is a slight increase in the weight-averaged molar mass followed by a decrease (*Figure 7A*). The increase in measured mass is due to the formation of C-P-C as WT-CaM binds to the preformed SKp/WT-CaM complex. The decrease in weight-averaged molar mass at higher concentrations of added WT-CaM is expected due to the rising levels of unbound WT-CaM, which is much smaller than C-P-C but dominates the composition of the solution.

Using preformed P-C with WT-CaM, we next added N-CaM or C-CaM. Instead of showing an increase in mass at low concentrations of added 'half-CaM', the measured weight average molar mass drops off with increasing mutant CaM (*Figure 7B*). The decrease in measured mass follows the predicted attenuation in weight average molar mass if the P-C complex remains intact, but the free N-CaM or C-CaM neither convincingly binds to SKp at another site nor replaces WT-CaM in the preformed complex. Taken together, *Figure 7A, B* shows that C-P-C requires two full-length CaMs.

## SKp binds one lobe of CaM if lobes are not joined

In *Figure 2*, combining N-CaM and C-CaM does not restore SK2 current. We wanted to know if unconnected lobes CaM can bind SKp at the same time. N-CaM and C-CaM individually bind SKp at high $Ca^{2+}$; however, a second unattached lobe does not measurably bind to SKp, that is, there is no C-P-C with the half-CaMs. If we mixed both N-CaM and C-CaM together, could the pair bind to SKp? In a cross-over CG-MALS experiment a sample with equimolar concentrations of N-CaM and C-CaM is varied against SKp. The results again indicate high-affinity binding of exactly one domain of CaM, not both, to SKp (*Figure 7C*). This experiment cannot distinguish how much of P-C is SKp bound to N-CaM vs. C-CaM. Unconnected N-CaM and C-CaM can only bind one at a time, suggesting that once bound to SKp, the other lobe is prevented from binding if it is not attached to the other lobe.

## CaM (E34Q) rescues SK2 current like WT-CaM, but NN-CaM and CC-CaM do not

Unconnected lobes of CaM do not activate SK, nor do they bind in the other's presence. We also wanted to know to what extent mutant full-length CaMs can activate SK. Previous work has shown that the CaM (E34Q) construct can substitute for WT-CaM to make functioning SK2 channels (*Keen et al., 1999*; *Li et al., 2009*). We confirmed those results here, while showing that fractional recovery of current is at least 50% of that of WT-CaM (*Figure 5A, B*). Due to variability in channel density and unknown occupancy by endogenous WT-CaM from the HEK cells, we are not able to confirm whether CaM (E34Q) rescues at a different rate or to a different saturation level than WT-CaM. Qualitatively, their abilities to rescue SK2 current are the same (*Figure 5C*).

NN-CaM and CC-CaM are approximately the same size as WT-CaM. If a CaM site on the C-terminus of SK can bind either lobe, an N-sensor view of CaM activation might suggest that at high $Ca^{2+}$, both lobes would be able to bind SK2, and perhaps, gate the channel. Neither NN-CaM nor CC-CaM activate SK2 convincingly (*Figure 5*). Thus, size is not the only factor, SK2 gating requires both an N- and a C-lobe.

## E to Q CaM constructs have different interactions with SKp than WT-CaM

CaM (E34Q) can activate SK2. Although SK2 activation suggests that CaM (E34Q) appears to function like WT-CaM, we currently do not know whether SK2 gating is the same with either CaM bound to it.

CG-MALS data show that WT-CaM forms three different stoichiometries at high $Ca^{2+}$ (*Figure 3A, K*). The profile for CaM (E34Q) looks different (*Figure 3D*). The characteristic 'M' shape is gone. Instead, there is a stronger peak at equimolar ratios of CaM (E34Q):SKp, and the right hump is absent. This can best be explained by a change in stoichiometries that form when SKp binds WT-CaM compared to CaM (E34Q). Whereas there is support for a P-C-P complex, as seen for WT-CaM, differences are seen at increasing molar ratios of CaM (E34Q). At equimolar ratios of WT-CaM:SKp, A 1/1 complex of CaM/SKp is observed with 28 kDa. However, the weight average molar mass at equimolar concentrations of CaM (E34Q):SKp is about 34 kDa (*Table 2*). The best solution to fit the data includes a small quantity of a 2/2 complex. We call this 2/2 complex the 2x-P-C (*Figure 3N*). At higher molar ratios of CaM (E34Q):SKp, there is no evidence for C-P-C, and in fact the weight average molar mass falls off quickly with increasing CaM (E34Q) arguing against a C-P-C with CaM (E34Q) (*Figure 3N*). In contrast, there are no obvious differences at trace $Ca^{2+}$ (*Figure 3F, I*).

We have shown that mutating only one N-lobe $Ca^{2+}$-binding site is sufficient to disrupt CaM binding to SK2 enough so that we can exchange CaMs in our rescue experiments. CaM (E12Q) forms high-affinity P-C with SKp at both low and high $Ca^{2+}$, but with weaker evidence for a P-C-P and no C-P-C (*Table 2*, *Figure 3E, J, O*).

We also used W432 spectra to evaluate the $Ca^{2+}$-dependent changes of these E to Q complexes. At trace $Ca^{2+}$, both CaM (E12Q) and CaM (E34Q) produce spectra similar to WT-CaM (*Figure 4G*). At 10 µM $Ca^{2+}$, the W432 spectrum with CaM (E12Q) bound is comparable in amplitude (<15%) and peak position (both 329 nm) to WT-CaM (*Figure 4H*). If $Ca^{2+}$ binding to the N-lobe is disrupted in E12Q as intended, the spectral overlap with WT-CaM strongly suggests that $Ca^{2+}$ binding to C-lobe is at least partially responsible for the spectral shift observed with WT-CaM (*Figure 4E*).

On the other hand, the spectrum of W432 with CaM (E34Q) bound has weaker $Ca^{2+}$ dependence (*Figure 4H*). $Ca^{2+}$ binding to the intact N-lobe of E34Q is not enough to produce a strong spectral change (≤3 nm shift and <20% from *Figure 4G, H*) that would overlap with the WT-CaM in $Ca^{2+}$ (≥4 nm and >30% from *Figure 4A, B*). 5 mM $Ca^{2+}$ does not shift the curve farther to smaller wavelengths indicating that higher $Ca^{2+}$ cannot drive a weaker interaction caused by the E to Q mutations (*Figure 4I*).

Summarizing the CaM (E34Q) results, (1) E34Q CaM rescues like WT-CaM, (2) CaM (E34Q) has different $Ca^{2+}$-dependent interactions with SKp than WT-CaM, and (3) $Ca^{2+}$ is less capable of altering the W432 environment when SKp is bound to CaM (E34Q) than when bound to WT-CaM. All these argue that the mutant CaM (E34Q) operates differently than WT-CaM to produce similar functional outcomes.

## NN-CaM and CC-CaM have Ca²⁺-dependent interactions with SKp

The double-lobe constructs were used to draw out more differences between the lobes. Neither construct can substitute for WT-CaM and restore channel activity (*Figure 5*). We used these constructs as tools to evaluate Ca²⁺ dependent binding to SKp.

N-CaM by itself has a low affinity for SKp at trace Ca²⁺ (*Figure 6A, C*). In contrast, NN-CaM binds with stronger affinity (p < 0.01) (*Figure 3B*, *Table 2*). Although data clearly display binding, the weight average molar mass is less than a P-C complex indicating a significant mole fraction of unbound protein in solution (*Figure 3G*). We observe that P-C-P may be present, but our confidence in higher-order complexes is low since multiple models for oligomers can produce a flatter shaped curve with weak binding. It's only when the weight average molar mass exceeds that of a 1/1 complex across molar ratios that models become more constrained for higher stoichiometries.

At high Ca²⁺, P-C-P and C-P-C both appear at lower mole fractions with NN-CaM than with WT-CaM (*Figure 3B, L*). The weight average molar masses exceed what would be expected by the P-C complex alone, so there we have some confidence that P-C-P and C-P-C are detected. At trace Ca²⁺, the weight-averaged masses are less than P-C, and there are more solutions involving different combinations of weaker affinities (*Figure 3B, G*). On the one hand, the similarity in the appearance of the data might argue that C-P-C exists even at trace Ca²⁺; however, we are more confident using a model with fewer variables and suggest that at a minimum the P-C is present with lesser amounts of C-P-C. The key observation is that NN-CaM binds stronger than N-CaM at trace Ca²⁺ and that there is a decreased Ca²⁺ sensitivity of complex formation compared with WT-CaM.

Except for geometric constraints, both lobes of CC-CaM are expected to bind SKp to make P-C-P. C-CaM by itself does not show Ca²⁺ sensitivity for stoichiometry formation, but the spectral data show that C-CaM binding to SKp is Ca²⁺ dependent (*Figure 4D, E*). Using CG-MALS, CC-CaM shows a small Ca²⁺-dependent change (*Figure 3C*). P-C-P shows up weaker at trace Ca²⁺ than at high Ca²⁺(-*Figure 3H, M*). Just like NN-CaM, CC-CaM shows a subtle but consistent hint of C-P-C at high Ca²⁺.

For a bimolecular interaction, for example, P-C, CG-MALS approaches a limit of detection for solving the $K_A$. For bimolecular fitting with no higher-order oligomers, log $K_A$ values above ~7.0 are not well constrained (*Table 2*). Other stoichiometries being present can broaden the range for the limit of detection. At high Ca²⁺, WT-CaM, CC-CaM, NN-CaM, and CaM (E34Q) all have features in the data that expand the range for solving for $K_A$ of P-C to higher values than 7.0 (*Figure 3A–D*). For example, the dip in the M-shape for WT-CaM can only be observed if there is a $K_A$ greater than 10. Even in this case there is some ambiguity about whether the limit was crossed, and it could be stronger yet. Thus, we limit our interpretations to be qualitative.

NN-CaM and CC-CaM binding confirm that both lobes retain Ca²⁺ sensitivity when bound to SKp and that each lobe has different properties. Both N- and C-lobes have Ca²⁺-sensitive interactions that SK2 decodes during function.

## Discussion
### The C-terminus of SK with bound CaM is a Ca²⁺ sensor for SK

CaM is tightly bound to SK regardless of Ca²⁺. It is required for SK to decode Ca²⁺ signals that vary in amplitude, location, and duration. CaM has an extremely versatile structure (*Figure 1C–F*). It can position its lobes, tune the Ca²⁺ sensitivity of its diverse partners, compete with other proteins, and have dynamic associations.

An early study narrowed the essential CaM modulatory domain to SK2's C-terminus, and there was no indication of another region of SK2 involved with CaM binding (*Xia et al., 1998*). Although recent work suggests that other domains of SK may play roles in gating and need to be considered in complete gating models (*Lee and MacKinnon, 2018*; *Nam et al., 2021*; *Figure 1A, B*), multiple lines of evidence show that the C-terminus is the critical Ca²⁺-sensing component in SK gating. Engineered mutations in the C-terminus affect trafficking and gating of channels in a heterologous system (*Keen et al., 1999*; *Lee et al., 2003*; *Nam et al., 2021*), whereas cells from different tissue control SK activity through splice variations in the C-terminus of SK (*Zhang et al., 2001*; *Zhang et al., 2012a*).

In addition to SK2's C-terminus, a second CaM-N-lobe-binding site has been shown on SK4 at the S4–S5 linker (*Lee and MacKinnon, 2018*). Additional studies of heterologously expressed SK2 channels show that mutations in either the S4–S5 linker or C-terminus can make SK2 more sensitive to

Ca$^{2+}$ (**Nam et al., 2021**; **Nam et al., 2017**; **Orfali et al., 2022**). Although there are mutations in S4–S5 that make SK2 gate at lower Ca$^{2+}$, more work is needed to determine whether the mutations alter an opened or closed state of SK2 so that it opens easier or if the mutations alter CaM interactions such that Ca$^{2+}$ binds the complex at lower concentrations. Binding measurements of CaM to other SK2 domains, such as the 4–5 linker, are difficult given the folding of the S4–S5 linker and have yet to be completed.

## Lessons from structures

If Ca$^{2+}$ affects C-lobe interactions at 10 µM concentrations, why then is Ca$^{2+}$ not seen as density in the C-lobe-binding sites of the structures solved at higher Ca$^{2+}$? High-resolution X-ray structures would be able to detect ions as strong electron densities, and there are methods to identify which metal ion is coordinated (**Echols et al., 2014**; **Harding, 2004**). The absence of Ca$^{2+}$ in the C-lobe is an experimental result that depends on the conditions used to arrive at the data. A major challenge in crystallography is getting a protein to form a crystal, which requires high protein concentrations in a somewhat arbitrarily determined medium. CaM co-crystallizes with SKp in acidic solutions that range from pH 4.6 to 5.6 at high ionic strength (~1 M Li$^{+}$) (**Schumacher et al., 2001**; **Zhang et al., 2012a**; **Zhang et al., 2013**). Both proton concentration and ionic strength alter the affinity of CaM for Ca$^{2+}$ (**Linse et al., 1991**; **Ogawa and Tanokura, 1984**; **Steiner et al., 1983**). Even so, in one instance the C-lobe of CaM still is Ca$^{2+}$ bound under acidic conditions when the SK peptide contains an additional three residues as a splice variant, showing that there are conditions at extreme pH (pH ~4.7) when the C-lobe is still Ca$^{2+}$ occupied (**Zhang et al., 2012a**). Interestingly, CaM was found with a C-shape in the original structure (**Figure 1E**), but with an S-shape with the splice variant (**Figure 1D**).

Recently, structures at low and at high Ca$^{2+}$ were determined for a full SK channel under different experimental conditions (**Lee and MacKinnon, 2018**): (1) it used the full-length channel as a tetramer with four bound CaMs with W-shapes (**Figure 1A–C**), (2) it used a more distantly related SK family member, the SK4 channel, (3) the media is at more physiological ionic strength (0.15 M) and at pH 8, and (4) electron densities were from cryogenic electron microscopy (cryo-EM). The structure obtained at high Ca$^{2+}$ shows that on CaM one C-lobe site is Ca$^{2+}$ bound and the other is Ca$^{2+}$ free. Cations are difficult to assign at worse than 3 Å resolution under conditions where electron densities are approachable experimentally (**Echols et al., 2014**; **Harding, 2004**). Both resolution limitations and the lack of anomalous scattering in modern electron cryo-EM make unambiguously assigning ions a challenge, but methods are emerging to locate and define bound metal ions (**Elad et al., 2017**). A vital question that remains is whether both C-lobe Ca$^{2+}$ are occupied at 10 µM, or if only one site is occupied as reported (**Lee and MacKinnon, 2018**). Our results do not distinguish occupancy of one vs. two Ca$^{2+}$ sites within a lobe. To our knowledge, there are no reports of lobe-specific Ca$^{2+}$-binding measurements of CaM-bound SK.

Even with a clearer picture of the architecture, high-resolution details remain elusive. Cryo-EM is an ensemble method that averages proteins across many states. Even though resolution is often the primary goal, this approach can explore different states (**Orlova and Saibil, 2010**; **Roh et al., 2017**; **Zhang et al., 2019**). Data are class averaged representations of thousands of particles. Class averaging minimizes differences while amplifying molecular similarities. Less frequent observed states are not necessarily minor functional states, and conversely more frequent observed states may not be functionally relevant. However, structures of a subset of states can generate testable hypotheses. Continued studies using cryo-EM on SK/CaM complexes would be valuable.

## Limits of the N-sensor hypothesis in SK gating

Across structures, W432 of SK2 binds the C-lobe of CaM (**Lee and MacKinnon, 2018**; **Schumacher et al., 2001**; **Zhang et al., 2012a**). The C-lobe was modeled with a putative Ca$^{2+}$ ion bound to one of its sites in the Ca$^{2+}$-bound cryo-EM structure. The similarity of the C-lobe structures at both trace and high Ca$^{2+}$ forms the basis for the N-sensor model. The basis for the N-sensor hypothesis is that the C-lobe is tethered independent of Ca$^{2+}$. In contrast to C-lobe binding, the N-lobe of CaM in subsets of cryo-EM class averaged data was observed in multiple conformations at trace Ca$^{2+}$ (**Lee and MacKinnon, 2018**). The flexibility of N-lobe binding could indicate different binding modes with SK in the absence of Ca$^{2+}$. Models from structures, however, are strongly biased toward what can be observed either in a crystal or through class averaging of thousands of molecules in cryo-EM. After particle

image class averaging, a small percentage (~40%) of total observed molecules is included in the final, highest 3.5 Å resolution structure of SK. There is an opportunity to learn what information is contained in the rest of the datasets about CaM binding.

The N-sensor hypothesis argues that the C-lobe has a weak $Ca^{2+}$ dependence, and therefore has weaker binding for $Ca^{2+}$. For binding, this would infer that the C-lobe would not alter the environment about W432 of SKp with added $Ca^{2+}$. Weaker $Ca^{2+}$ binding to the C-lobe is not consistent with our data.

At minimally saturating $Ca^{2+}$, the spectra for CaM (E34Q) should bind SK like WT-CaM and CaM (E12Q) should appear like WT-CaM does at trace $Ca^{2+}$. Our data show the opposite. Our results are consistent with what is expected if both lobes bind $Ca^{2+}$ and better support the NC-sensor hypothesis. CaM has four $Ca^{2+}$-binding sites, and the NC-sensor hypothesis is more intuitive from an energetic scheme that includes all ligand-binding sites.

The N-sensor hypothesis for SK gating by WT-CaM (*Figure 1*) is not supported by binding data, but the mutant CaMs show results that could have been considered as consistent with the N-sensor hypothesis. CaM (E34Q), whether overexpressed (*Keen et al., 1999*) or reconstituted (*Li et al., 2009*; *Figure 5*), activates SK2. If the E to Q mutations only alter $Ca^{2+}$ binding to CaM, the CaM (E34Q) data suggest that $Ca^{2+}$ binding to N-lobe is required for SK gating, but not to the C-lobe. The assumption was that CaM (E34Q) activates SK2 the same as WT-CaM. However, our protein-binding data challenges that assumption. CaM (E34Q) and WT-CaM do not interact with SK in the same way and CaM (E34Q) clearly has a different binding mode.

## SK gating requires both lobes of CaM

How does $Ca^{2+}$ open SK channels? The answer is maybe it doesn't; $Ca^{2+}$ just keeps SK from closing. *Hirschberg et al., 1998* used single-channel recordings of SK2 to show that the SK2 has many closed and open states. A key finding was that only a closed state with long durations is unambiguously $Ca^{2+}$ sensitive; none of the open states or short duration closed states are $Ca^{2+}$ sensitive. This observation sets a premise for SK gating: $Ca^{2+}$ must do work to prevent SK from visiting the closed state that has the longest dwell time. When SK is activated, this essentially means that long closed times are diminished. More binding data, electrophysiology, and structures are essential to learn how SK traverses different states in its response to $Ca^{2+}$.

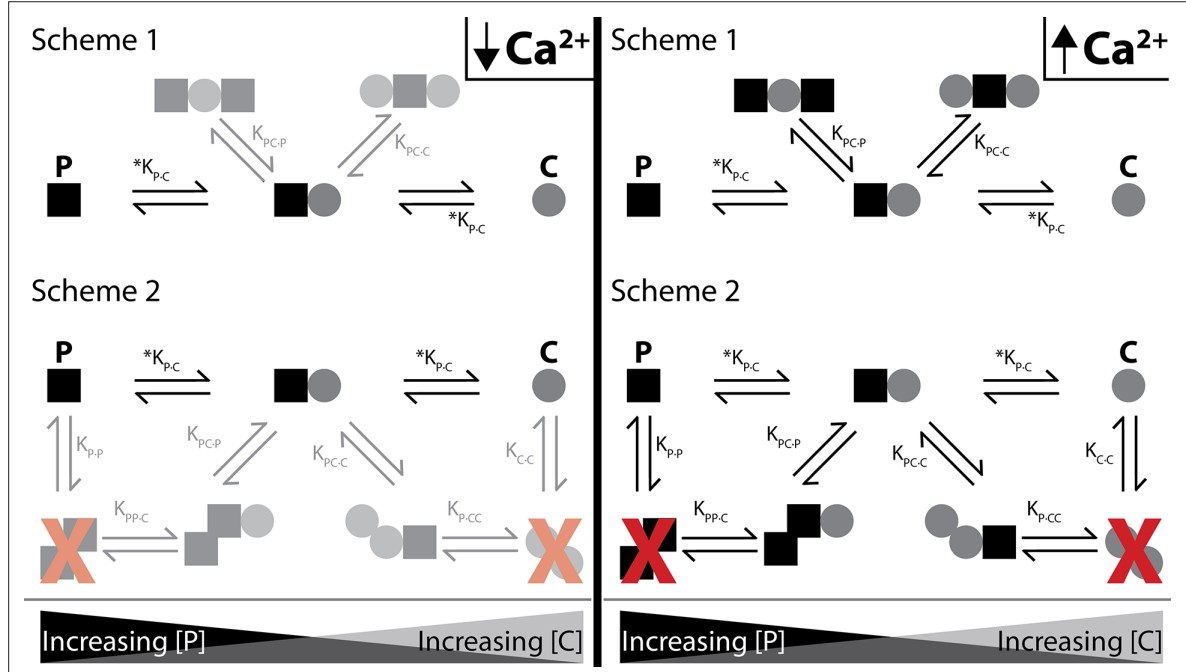

**Figure 8.** Shematics showing different configurations of stoichiometries between WT-CaM and SKp. SKp is represented as a black square; CaM is represented as a gray circle. Different stoichiometries are favored at different molar ratios of peptide and CaM in solution. $Ca^{2+}$ allows peptide and CaM to bind with additional stoichiometries.

SK gating by CaM requires various properties of CaM. From our electrophysiology (*Figure 5*) we find that the following are necessary for CaM to gate SK: (1) CaM must have two joined lobes. (2) CaM must have both an N- and a C-lobe. (3) The N-lobe needs intact $Ca^{2+}$ sites. We infer that CaM with mutations in C-lobe, that is, CaM (E34Q), can activate via an alternate mechanism.

Our binding data reveal that CaM has dynamic associations with a peptide in response to $Ca^{2+}$. One limitation of our study is that a peptide is free to bind CaM in configurations that are less constrained than it would be in a membrane-bound channel. Further binding data with CaM to full channels are needed. Regardless, the peptide-binding data still inform that $Ca^{2+}$ alters binding as expected for a sensor. We learn from the C-P-C configuration that $Ca^{2+}$ opens a site on SKp that is different than when there is no $Ca^{2+}$ present. In the context of a full channel, there might not be room for CaM molecules to associate (*Lee and MacKinnon, 2018*), but in terms of binding it means that at high $Ca^{2+}$, CaM has more ways to interact with SK.

C-P-C binds strongest with WT-CaM and only in $Ca^{2+}$ (*Figure 8*). The double-lobe mutants CC-CaM and NN-CaM show a suggestion of C-P-C, but with WT-CaM C-P-C is clearly stronger. With half-CaMs, neither N-CaM nor C-CaM form C-P-C, although either one binds SKp with high affinity. If the N- and C-lobes bind at different specialized sites, there would be an expectation that both the N-CaM and C-CaM can bind SKp together, but they do not (*Figure 7C*). However, C-P-C requires both an N- and a C-lobe. One alternative that has not yet been tested is that the 'linker' between lobes contains the necessary requirements for C-P-C formation. N-CaM and C-CaM each possess part of the linker and overlap at residue positions 78–80 (DTD). The presence of part of the linker on each construct somewhat tests the hypothesis that the linker may need to be present to form C-P-C, although, because there is not a full and intact continuous linker present, half-CaM's with variable linkers would be needed for testing whether the linker is required for C-P-C formation.

For C-P-C, both lobes of both CaMs exchange near a common binding site that CaM only senses in the presence of $Ca^{2+}$. There are two possible arrangements that our data cannot distinguish that we present as scheme 1 and scheme 2 (*Figure 8*). In scheme 1, two CaMs bind directly to SKp and have no association with the other CaM. In Scheme 2, SKp binds to a CaM oligomer. We favor scheme 1 because our binding measurements have not detected any CaM dimers in our solution measurements.

Testing these constraints requires having probes that can measure the N- and C-lobe positions on the peptide and the conformation of the P-C complex. Our hypothesis predicts multiple conformations of the P-C complex to account for the binding behavior of WT-CaM. Data that show distributions of different conformations would support this conjecture. In contrast, CaM (E34Q) binding to SKp potentially can be explained with fewer states.

For P-C-P, we speculate that both lobes of a single CaM molecule participate with binding to both SKp peptides, possibly in a more globular form but not necessarily in a 'bridging' conformation (*Figure 8*). Indeed, CaM binds many targets with an O-shape (*Figure 1F*), except that the O-shape has not yet been observed in structures with SK. We do not detect dimers of SKp during our measurements using purified peptide, yet we note that impurities in the solution may cause the peptide to precipitate. Since we do not observe SKp dimers, we favor scheme 1 in *Figure 8* to describe P-C-P.

Finally, at trace $Ca^{2+}$, we only consider a strong P-C signature as convincing (*Figure 8*). Further investigation is needed to find out whether P-C-P exists at trace $Ca^{2+}$ and how the weaker binding would fit into a gating model. In our current model, both lobes of CaM interact with SK at trace $Ca^{2+}$ in the P-C state.

## Predictions of SK gating from CaM-binding data

A more general model is needed to describe how SK decodes $Ca^{2+}$ binding to both lobes of CaM. The 'NC-sensor hypothesis' proposes that binding of $Ca^{2+}$ to both lobes of CaM in complex with SKp are energetically coupled processes, that is, the entire CaM molecule works as a unit and *not* as a sum of its parts. Although both lobes have unique qualities as they do in the N-sensor hypothesis, binding data suggest that they work as a unit by communicating through SK. The NC-sensor hypothesis has testable predictions. Both WT-CaM and CaM (E34Q) activate SK, but since one lobe of CaM (E34Q) is less $Ca^{2+}$ sensitive, SK must have different dwell times in the opened or closed states. Single-channel recordings of SK current can probe different activation states of SK. A careful study of distributions of SK states would distinguish whether different opened and closed states are prevalent if CaM (E34Q) is bound instead of WT-CaM.

Single-channel recordings of SK2 show that in trace $Ca^{2+}$ the open probability is dominated by a long-lived closed state (*Hirschberg et al., 1998*). Rare bursts of activity show openings that are marked with flickers to closed states. At high $Ca^{2+}$, the channel is open about 80% of the time. The 'open' channel still has frequent flickers to a closed state. The increase in open probability seems to be entirely due to a drastic reduction in the long-lived closed state and not an increase in time spent in the open state, that is, there is not a decrease in 'flickering'. Thus, at high $Ca^{2+}$, SK2 shows extremely dynamic behavior. At equilibrium, a distribution of different states of SK2 should be expected. Whether CaM (E34Q) is bound is expected to modify this behavior.

More structures of SK with bound CaM captured in identifiable closed and open intermediates would help to interpret how CaM alters SK activity. We know of at least three closed and two open states for SK activity from physiology (*Hirschberg et al., 1998*). Our stoichiometry data show that SK has multiple ways to bind CaM, and there is ample structural support displaying how flexible CaM is when bound to targets. Our binding and functional work are only edge cases for concentrations and there is much more work to be done to solve what the protein experiences dynamically during gating.

## A generalized hypothesis for SK gating

CaM binding to peptide may be different from how it binds to a full-length protein such as a channel, especially if another domain such as the 3–4 linker interacts with a lobe of CaM. This is an extraordinarily difficult question to answer with just one binding study, yet incorrect aspects of models can quickly be eliminated. We observe that the C-lobe of CaM is calcium sensitive when binding to a peptide. How can the full-length channel binding to the C-lobe be $Ca^{2+}$ insensitive? It would require that the full-length channel blocks the $Ca^{2+}$ C-lobe configuration while preserving the $Ca^{2+}$-free configuration, but this has not been demonstrated. Considering that the channel has at least three closed states and two open states, C-lobe $Ca^{2+}$ sensing is highly likely to be important during gating. From current structure analyses, we have at best an ensemble view of two out of five channel/CaM states.

Although the N-sensor hypothesis is elegant, the binding data show that a more complex model is likely necessary to truly represent the mechanism for SK gating by WT-CaM (*Figures 1G and 8*). Both lobes of CaM are energetically coupled through binding to SK. The complex of SK and CaM decodes a $Ca^{2+}$ signal, with both lobes in complex with SK acting as sensors. The C-lobe of CaM is a $Ca^{2+}$ sensor; we show that it is not insensitive to $Ca^{2+}$ as described in the N-sensor hypothesis. We propose that the mechanism for the NC-sensor hypothesis requires solving energetic values for $Ca^{2+}$ binding to four ligand sites of CaM, multiple CaMs binding to SK, and SK open probability at different liganded states. The challenge of solving energetic states is far greater and more general in the NC-sensor hypothesis, but it provides a realistic starting point for solving the mechanism of SK activation.

## Methods
### Materials and constructs

The mammalian WT-CaM protein sequence (NCBI: NP_059022) expression clone from rat DNA (Rattus norvegicus calmodulin 2 (Calm2) NCBI: NM_017326.3) was provided by S. Hamilton (Baylor College of Medicine, Houston, TX), which we cloned into the Novagen pET21a vector (Merck KGaA, Darmstadt, Germany) for expression in BL21(DE3) bacteria (New England Biolabs (NEB), Ipswich, MA). Double lobe CaM DNA constructs, NN-CaM and CC-CaM were codon optimized for bacterial expression and synthesized by Genewiz (South Plainfield, NJ). J. Adelman (Vollum Institute, Portland, OR) provided the clone for the Rattus norvegicus SKp (residues 396–487 rat KCa2.2, UniProt accession P70604.1) of SK2, and was cloned without a tag as described previously (*Halling et al., 2014*). The human protein sequence of SK2 is identical in this region. Reagents used for buffers were minimal American Chemical Society quality and purchased from Sigma-Aldrich unless otherwise indicated. All solutions were made with 18.2 MΩ water from a Milli-Q Integral 5 System (Burlington, MA).

### Cloning of proteins for expression

The N-CaM construct was designed by cloning in a stop codon (TAA) into the WT-CaM construct in the pET21a vector using standard PCR mutagenesis after residue D80 of the protein sequence using primers ordered from Sigma-Aldridge (Merck). C-CaM was cloned using PCR amplification of the C-terminus of WT-CaM with added flanks of a 5′ NdeI overhang and a 3′ BamHI overhang and ligated

into pET21a vector (Merck). Insertion of PCR product into pET21a was achieved with standard protocols (NEB).

The CaM constructs, we tested fall into four categories. (1) WT-CaM is used as our standard. (2) Isolated CaM domains, that is, isolated N-terminal domain (N-CaM) or C-terminal domain (C-CaM), were used to determine if Ca²⁺ binding to an isolated domain of CaM by itself could activate SK2. In some experiments, we used N-CaM and C-CaM in the same mixture to see if the parts of CaM could restore SK2 activity without both lobes being joined. (3) Double-lobe CaMs that have repeated domains were used to determine if the domains of CaM have compensating roles in SK2 activation. We made CaM with two N-lobes but no C-lobe (NN-CaM) and CaM with two C-lobes but no N-lobe (CC-CaM). (4) Finally, we used full-length CaM mutants that are deficient for Ca²⁺ binding to one domain, but the opposite domain has normal Ca²⁺ binding. The CaM (E12Q) mutant CaM has glutamines (Q) in place of glutamates (E) and positions 31 and 67 in the N-lobe of CaM. The E to Q mutations in the N-lobe were reported to disrupt Ca²⁺ binding to sites 1 and 2 (*Mukherjea et al., 1996*). Similarly, the CaM (E34Q) mutant, which has E to Q mutations at positions 104 and 140, was reported to disrupt binding to sites 3 and 4 in the C-lobe.

For CaM protein sequence, the N-terminal methionine gets cleaved in the final protein product and the next alanine is acetylated (*Sasagawa et al., 1982*). In our protein constructs, the N-terminal methionine, which is subject to cleavage during expression, is referred to as position M0. The N-CaM protein sequence (residues M0-D80 of WT-CaM) is:

> MADQLTEEQIAEFKEAFSLFDKDGDGTITTKELGTVMRSLGQNPTEAELQDMINEVDADGNGTI
> DFPEFLTMMARKMKDTD-
> MADQLTEEQIAEFKEAFSLFDKDGDGTITTKELGTVMRSLGQNPTEAELQDMINEVDADGNGTI
> DFPEFLTMMARKMKDTD-
> The C-CaM protein sequence (M76-K148) is
> MKDTDSEEEIREAFRVFDKDGNGYISAAELRHVMTNLGEKLTDEEVDEMIREADIDGDGQVNYE
> EFVQMMTAK-
> MKDTDSEEEIREAFRVFDKDGNGYISAAELRHVMTNLGEKLTDEEVDEMIREADIDGDGQVNYE
> EFVQMMTAK-

NN-CaM and CC-CaM were purchased as described in materials and arrived in pUC57-Kanamycin vector. The gene sequences were flanked by restriction enzyme sites for 5′ NdeI and 3′ EcoRI (NEB). Genes were cloned into pET21a using standard protocols from NEB. Due to a need for a start codon, the CC-CaM positions 1 and 2 (residues MA) do not align with WT-CaM. Residues D78–D81 of WT-CaM were retained as the flexible part of the linker in both constructs of NN-CaM and CC-CaM. Some effort was made to make the linkers of NN-CaM and CC-CaM of similar length.

> The NN-CaM protein sequence (residues M0-D81, Q8-K77) is MADQLTEEQIAEFKEAFSLF
> DKDGDGTITTKELGTVMRSLGQNPTEAELQDMINEVDADGNGTIDFPEFLTMMARKMKDTDQIA
> EFKEAFSLFDKDGDGTITTKELGTVMRSLGQNPTEAELQDMINEVDADGNGTIDFPEFLTMMARK
> The NN-CaM protein sequence (residues M0-D81, Q8-K77) is MADQLTEEQIAEFKEAFSLF
> DKDGDGTITTKELGTVMRSLGQNPTEAELQDMINEVDADGNGTIDFPEFLTMMARKMKDT
> DQIAEFKEAFSLFDKDGDGTITTKELGTVMRSLGQNPTEAELQDMINEVDADGNGTIDFPEFLT
> MMARK
> The NN-CaM protein sequence (residues M0-D81, Q8-K77) is MADQLTEEQIAEFKEAFSLF
> DKDGDGTITTKELGTVMRSLGQNPTEAELQDMINEVDADGNGTIDFPEFLTMMARKMKDTDQIA
> EFKEAFSLFDKDGDGTITTKELGTVMRSLGQNPTEAELQDMINEVDADGNGTIDFPEFLTMMARK
> The CC-CaM protein sequence (residues S81-K148, D79-K148) is
> MASEEEIREAFRVFDKDGNGYISAAELRHVMTNLGEKLTDEEVDEMIREADIDGDGQVNYEEFV
> QMMTAKDTDSEEEIREAFRVFDKDGNGYISAAELRHVMTNLGEKLTDEEVDEMIREADIDGDGQ
> VNYEEFVQMMTAK
> MASEEEIREAFRVFDKDGNGYISAAELRHVMTNLGEKLTDEEVDEMIREADIDGDGQVNYEEFV
> QMMTAKDTDSEEEIREAFRVFDKDGNGYISAAELRHVMTNLGEKLTDEEVDEMIREADIDGDGQ
> VNYEEFVQMMTAK
> MASEEEIREAFRVFDKDGNGYISAAELRHVMTNLGEKLTDEEVDEMIREADIDGDGQVNYEEFV
> QMMTAKDTDSEEEIREAFRVFDKDGNGYISAAELRHVMTNLGEKLTDEEVDEMIREADIDGDGQ
> VNYEEFVQMMTAK

## Protein preparation

A single colony of a protein construct in BL21(DE3) cells is selected with a toothpick and put into 20 ml of Luria Broth (Fisher Scientific Company LLC, Pittsburg, PA) with appropriate antibiotic for the construct, either carbenicillin or kanamycin. These 'starter cultures' are grown for 7–8 hr, but no more than 8 hr, at 37°C and stored in the fridge overnight. The next day, the starter culture is used to inoculate 750 ml LB in a 2 l flask and the cultures are grown at 37°C in an orbital shaker to an optical density of 0.6 absorbance units at a wavelength of 600 nm. The temperature of the shaker is either kept at 37°C for SKp protein or changed to 25°C for CaM constructs. Cultures are then induced with 0.5 mM isopropyl β-D-1-thiogalactopyranoside and induced for 2–3 hr for SKp or for 16–18 hr for CaM constructs. Proteins were purified using established procedures (*Halling et al., 2014*). Only proteins that migrate as a single band on a 20% Laemmli gel, that is, negligible contaminants and degradation in sample, are used in experiments.

## Cell lines: HEK293 cell maintenance and transfection

293 [HEK-293] (ATCC CRL-1573) cells were STR tested and are negative for mycoplasma (ATCC, Manassas, VA). 293 cells were grown and passaged according to standard procedures. Cells were cultured at 37°C in a 5% $CO_2$ atmosphere in Dulbecco's modified Eagle's medium without sodium pyruvate and with L-glutamine, 1% penicillin/streptomycin, 1% GlutaMAX, and 10% fetal bovine serum (Thermo Fisher Scientific). Cell lines were split with trypsin/EDTA in Hanks' balanced salt solution (Thermo Fisher Scientific) up to 20–25 cycles. For experimental electrophysiological recordings, cells were transiently cotransfected (Lipofectamine 3000; Invitrogen) with the rat SK2 channel and the E1Q calmodulin mutant. For control experiments, cells were transfected with either the rat SK2 channel and wild-type calmodulin, or the rat SK2 channel alone. Enhanced green fluorescent protein was also cotransfected as a marker in all cases. Electrophysiological recordings were done 16–72 hr after transfection.

## Patch-clamp recordings

Patch-clamp recordings of SK currents were performed at room temperature (22–24°C) using an inside-out configuration. Recording electrodes were fabricated with borosilicate glass pipets (A-M Systems) and pressure/fire-polished before use. Open electrode resistance was 2–4 MΩ in the bath solution. An Axopatch 200 A amplifier (Molecular Devices San Jose, CA) coupled with an ITC-18 interface (HEKA) was used to record data. Acquisition software (PATCHMASTER; HEKA; Lambrecht, Germany) was used to sample data at 50 kHz, which were low-pass filtered at 1 kHz using the built-in filter of the amplifier. SK currents were elicited by stepping to a voltage of −60 mV for 1 s every 3 s until recording was paused for solution changes.

For all recordings, the extracellular (pipette) solution contained the following: 2 mM KCl, 136 mM KOH, 20 mM HEPES (4-(2-Hydroxyethyl)piperazine-1-ethanesulfonic acid, N-(2-Hydroxyethyl)piperazine-N'-(2-ethanesulfonic acid)), and 2 mM $MgCl_2$, adjusted to pH 7.2 with $MeSO_3H$. The intracellular (bath) solution contained the following: 6 mM KCl, 132 mM KOH, and 20 mM HEPES, adjusted to pH 7.2 with $MeSO_3H$. $Ca^{2+}$ chelators were used in the intracellular solution to control free $Ca^{2+}$ and (+)-(Crown-6)-2,3,11,12-tetracarboxylic acid (Synquest Laboratories) was used to chelate any contaminating barium to prevent barium block of channels. 3.06 mM $CaCl_2$ and 5 mM HEDTA (N-Carboxymethyl-N'-(2-hydroxyethyl)-N,N'-ethylenediglycine) were added to the internal solution for an estimated 10 μM free $Ca^{2+}$ (MaxChelator) and was verified by measurement with a $Ca^{2+}$-sensitive electrode. Solution with 5 mM EGTA and no added $Ca^{2+}$ was considered as nominally $Ca^{2+}$ free and was used to estimate leak current levels. All solutions containing calmodulin protein were made using 10 μM free $Ca^{2+}$ internal solution and the concentration of either WT-CaM or non-WT constructs was 20 μM. For solution changes, a pressurized SmartSquirt (Automate Scientific) micro-perfusion system was used. When switching between $Ca^{2+}$ solutions with or without exogenous calmodulin protein, the patch pipette was moved in close proximity to the SmartSquirt perfusion pencil (250 μM opening) which was immersed in the bath chamber.

The experimental protocol was the following: patches were excised from cells immersed in a bath chamber containing 10 μM free $Ca^{2+}$. Using the SmartSquirt, the solution was changed between $Ca^{2+}$ free (to estimate leak and confirm that current was $Ca^{2+}$ dependent) and 10 μM free $Ca^{2+}$ until the current was stable across solution changes. Then, the exogenous CaM protein was perfused on using

the SmartSquirt, which varied in the amount of time from patch to patch. Early experiments had more variability than later experiments, in the time CaM protein was perfused on, due to the need to determine how long it took to see a measurable effect on current levels. If it was an experiment where two CaM constructs were being applied to the patch, the non-WT-CaM always were applied first, and the solution was changed to $Ca^{2+}$-free solution and then back to 10 µM free $Ca^{2+}$ before finally perfusing on WT-CaM. All measurements of current change, in experiments where both non-WT-CaM and WT-CaM were applied to the patch, were done over the same amount of time for both CaM constructs. However, the amount of time each construct was being applied varied from patch to patch due to the need to conserve exogenous protein and the variability of patch lifetime.

In some experiments, where SK current ran up slightly during the course of the experiment, a drift correction was performed by fitting a line to the drift and subtracting it from the average current level.

## Light scattering

Stock protein samples were centrifuged at 17 kg prior to dilution into the working buffer. Protein concentrations were measured in a cuvette with 1-cm path length and using a UV-2600 UV-spectrophotometer (Shimadzu Scientific Instruments, Inc, Columbia, MD) to collect protein spectra. The average absorbance was subtracted by baseline averages from 330 to 350 nm. Protein absorbance was converted to concentrations using predicted extinction coefficients ($\varepsilon$) based on numbers of tyrosines and tryptophans using either ProtParam (*Gasteiger et al., 2005*) or phenylalanines as follows. Free phenylalanine $\varepsilon$ was determined by weighing desiccated L-phenylalanine with milligram accuracy, dissolving into 1 ml of 18 MΩ water and collecting spectra from 350 to 250 nm. The ε was computed by the relation $\varepsilon = A/Lc$, where $A$ = absorbance, $L$ is the path length = 1 cm, $c$ is the molar concentration of L-phenylalanine. L-Phenylalanine has a spectral maximum at 258 nm, and $\varepsilon$ was measured at this wavelength for phenylalanine. We assume a linear relation of the number of phenylalanines in the protein with total $\varepsilon$. The values we used for $\varepsilon$ in units of $M^{-1}$ $cm^{-1}$ (at given wavelength) are $\varepsilon$ = 6990 SKp (280 nm), 3020 WT-CaM (277 nm), 975 N-CaM (258 nm), 3020 C-CaM (277 nm), 1950 NN-CaM (258 nm), 5960 CC-CaM (277 nm), and 3020 for CaM (E34Q)- and CaM (E12Q) (277 nm).

Working solutions for light scattering were filtered through Whatman 0.02-µm aluminum oxide filters (GE Healthcare Bio-Sciences, Pittsburgh, PA). All polystyrene bottles (Nalgene, Rochester, NY) for solution storage were rinsed 10 times in 18.2 MΩ water. Buffer reagents were purchased from Sigma-Aldrich (subsidiary of Merck). Azide was added as a preservative. Low $Ca^{2+}$ (<5 nM free $Ca^{2+}$) buffer contained 100 mM NaCl, 10 mM HEPES, 5 mM EGTA, 0.02% $NaN_3$ pH 7.0. High $Ca^{2+}$ buffer contained 100 mM NaCl, 10 mM HEPES, 5 mM $CaCl_2$, 0.02% $NaN_3$ pH 7.0. Some controls were performed in 10 µM free $Ca^{2+}$ buffer which contained 100 mM NaCl, 10 mM HEPES, 5 mM HEDTA, 2.8 mM total $CaCl_2$, 0.02% $NaN_3$ pH 7.0. The final free $Ca^{2+}$ concentration was verified with an Orion 93-20 calcium half-electrode (Thermo Electron Corporation, Waltham, MA).

Stock protein solutions were diluted into a working buffer, 0.096 mg/ml SKp and 0.09 mg/ml for any CaM construct. Diluted samples were again centrifuged for 6.5 kg for 20 min. Composition-gradient multi-angle light-scattering experiments were carried out as previously described (*Halling et al., 2014*) on a Calypso-II (Wyatt Technology, Santa Barbara, CA) pump delivery system to a Dawn Heleos-II (Wyatt) multi-angle 18 detector system held at 25°C. An inline UV monitor 170 S was used to monitor protein absorbance at 280 nm (Bio-Rad, Hercules, CA). Monomeric protein concentration was evaluated in each experiment by stepping concentrations and monitoring UV absorbance and light-scattering intensity. The molar masses of monomeric SKp and CaMs are known and used to determine if any small adjustments were needed to correct SKp or CaM concentrations. After SKp and CaM concentrations are verified, analysis of the 'cross-over' phase of the experiment was carried out using Calypso software (Wyatt) as described previously (*Halling et al., 2014*), except that we determined that fitting for 'incompetent fraction' was unnecessary. For small molecules relative to the light wavelength and at low concentrations, light-scattering signals are not angular dependent and corrections from virial coefficients at high concentrations are negligible. Under these simplified conditions, the weight-averaged molar mass of all particles in solution, $M_z = R(\theta,c)/(P(\theta) \cdot K^* \cdot c)$, is directly computed (*Wyatt, 1993*). $R(\theta,c)$ is the excess Rayleigh ratio of the light-scattering caused by the dissolved protein in a solvent. It is the ratio of protein plus solvent to solvent alone. For small molecules, $R(\theta,c)$ is virtually angle independent ($P(\theta) \approx 1$), but changes as a function of protein concentration, $c$. The value $K^* = 4\pi^2(dn/dc)^2 n_o^2/(N_A \lambda_o^2)$ is constant. $N_A$ is Avagadro's number, $\lambda_o$ is the laser

wavelength in a vacuum ($\lambda_o$=660 nm), $n_o$ is the solvent refractive index ($n_o \approx 1.33$), and d$n$/d$c$ is the proteins refractive index increment. We measured the d$n$/d$c$ of CaM to be 0.186 on an Optilab rEX refractometer (Wyatt) using Astra 7.1.1.3 software, which is consistent with published experimental (0.185) and computed values (0.190) for proteins (*Khago et al., 2018*; *Zhao et al., 2011*).

Analyses on light-scattering samples were performed in the software Calypso-II Version 2.2.4.32 (Wyatt). Observed light-scattering intensity at different molar ratios of protein and ligand was used to model the proportions of free vs. bound proteins in the sample. The total concentrations of added protein are known and verified before and after experiments as part of the method. Their monomeric molar masses are known from sequence and were verified by mass spectrometry on an AB Voyager-DE PRO MALDI-TOF instrument (Applied Biosystems now Thermo Fisher). Stoichiometries are assumed to be comprised of known molar masses of SKp and CaM. Free and bound protein concentrations were modeled across molar ratios as previously described (*Halling et al., 2014*). The concentrations of free protein and complexed protein were fitted to $M_z$ by Levenberg–Marquardt fitting of the equilibrium association constants, $K_A$, for each complex.

## Tryptophan emission spectra

SKp contains a single, native tryptophan at position 432, whereas CaM has no tryptophan. Between the two proteins, there are three tyrosines, but tryptophan fluorescence is dominant over tyrosine. W432 has been shown to contact the C-lobe of CaM in multiple structures (*Nam et al., 2017*; *Schumacher et al., 2001*; *Zhang et al., 2012a*; *Zhang et al., 2013*), and its spectra have been used to assess WT-CaM binding to SKp (*Keen et al., 1999*). Because of its position and potential role in forming a CaM-binding site, W432 spectra could be used as a probe in assessing CaM binding under more conditions.

1 µM SKp (W432) plus/minus 1 µM CaM (no tryptophan) were dissolved in the same working solutions used for CG-MALS. Fluorescence spectra were collected in a 1-cm pathlength Spectrosil quartz cuvette (Starna, Atascadero, CA) on a QM-400 PTI spectrofluorometer (Horiba Scientific, Kyoto, Japan) containing a 75-W xenon arc lamp and an R928 side-on photomultiplier (PMT) detector (Hamamatsu Photonics K.K., Shizuoka, Japan). The light path included an excitation monochromator, a 2-nm bandwidth excitation slit, a cuvette with sample, a 305-nm cut-on emission filter, a 1-nm bandwidth emission slit, emission monochromator, and PMT. Fluorescence was excited at 265 nm and its emission was scanned in 1 nm steps from 315 to 400 nm with 1-s signal integrations. All scans are buffer background subtracted. CaMs, including those with tyrosines, were measured for controls. Averaged spectra were subjected to binomial smoothing using Igor Version 5.0.5.7 (Wavemetrics Inc, Lake Oswego, OR).

# Additional information

### Competing interests
Richard W Aldrich: Senior editor, *eLife*. The other authors declare that no competing interests exist.

### Funding

| Funder | Grant reference number | Author |
|---|---|---|
| National Institute of Neurological Disorders and Stroke | 5R01NS077821-10 | Richard W Aldrich |
| National Institute of General Medical Sciences | 5R01GM127332-03 | Richard W Aldrich |

The funders had no role in study design, data collection, and interpretation, or the decision to submit the work for publication.

### Author contributions
David B Halling, Conceptualization, Resources, Data curation, Formal analysis, Supervision, Funding acquisition, Validation, Investigation, Visualization, Methodology, Writing - original draft, Project

administration, Writing - review and editing; Ashley E Philpo, Conceptualization, Data curation, Formal analysis, Validation, Investigation, Visualization, Methodology, Writing - original draft, Project administration, Writing - review and editing; Richard W Aldrich, Conceptualization, Resources, Formal analysis, Supervision, Funding acquisition, Validation, Investigation, Visualization, Methodology, Writing - original draft, Project administration, Writing - review and editing

### Author ORCIDs
David B Halling ![ORCID] http://orcid.org/0000-0003-0921-8738
Richard W Aldrich ![ORCID] http://orcid.org/0000-0001-7254-5876

### Decision letter and Author response
Decision letter https://doi.org/10.7554/eLife.81303.sa1
Author response https://doi.org/10.7554/eLife.81303.sa2

---

## Additional files

### Supplementary files
• MDAR checklist

### Data availability
All data generated or analyzed during this study are included in the manuscript.

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
