## [Editor Report]

This manuscript provides compelling evidence that in response to calcium, both lobes of the protein calmodulin change their interaction with a domain of a potassium channel. These findings provide valuable information about the molecular mechanics by which calcium binding is transduced to channels. Calmodulin is important for many biological processes and this work is expected to be of interest to researchers studying biophysics, protein conformational change, calcium signaling, and general physiology.

---

## [Decision Letter]

**Decision letter after peer review:**

Thank you for submitting your article "Calcium dependence of both lobes of calmodulin is involved in SK channel gating" for consideration by *eLife*. Your article has been reviewed by 3 peer reviewers, including Jon T Sack as the Reviewing Editor and Reviewer #1, and the evaluation has been overseen by a Reviewing Editor and Kenton Swartz as the Senior Editor.

Essential revisions:

1) Please make it abundantly clear to readers (e.g., in the Abstract and Discussion) that while the results suggested that ca^2+^ binding to the C-lobe causes a conformational change, electrophysiological experiments did not conclusively show that ca^2+^ binding to the C-lobe of CaM impacts SK channel gating. In the Abstract and end of the Discussion, the wording seems to project conclusions from soluble SKp onto SK conductance activation without a disclaimer. It might help to expound on the limitations of projecting results with SKp onto gating of intact SK channels.

2) The conclusion that the X-C CaM – SKp complex changes conformation in ca^2+^ is important, but difficult to see in Figure 4. Please, provide a quantitative comparison of the data in Figure 4 which leads to important conclusions, so that readers who may struggle to overlay emission spectra in their heads can benefit.

3) Please address each reviewer's concern with changes to the manuscript, or explain why not in the rebuttal.

*Reviewer #1 (Recommendations for the authors):*

Despite trying hard to find flaws in the logic of interpretation, I've come up empty-handed. The interpretations seem logically watertight, with the exception of a few wording choices in the abstract and final paragraph.

My biggest suggestion is to make the identity of SKp and the limitations of using only the C-terminal peptide more obvious in Figure 1 and the introduction.

One interesting finding that might be worth discussing further is that the change log KA was similar for WT-CaM and N-CaM (Table 2). Maybe this is expected, but could this be interpreted as evidence that the energy of ca^2+^-induced binding of N-CaM is sufficient to account for the binding of WT-CaM?

D->A mutations in the C-lobe 3,4 sites decrease ca^2+^ sensitivity and Hill coefficient of SK activation (Xia et al., 1998). It could be helpful to discuss whether this prior finding provides evidence that the C-lobe ca^2+^ sites are important for binding or efficacy.

Specific comments:

Abstract

"widely accepted": suggest removing social speculation.

"both lobes bound to SK are ca^2+^ sensitive": suggest changing "SK" to "SKp".

Introduction

Page 1

"…which results…" → which can result?

Results

Page 5

"Intrinsic tryptophan fluorescence of SKp confirms calcium sensitivity in both lobes of calmodulin."

The results showing this are not discussed in this section. Only WT CaM is discussed here.

Page 5

…peak is shifted towards shorter wavelengths (329 nm)…

please provide a quantitative estimate of shifts and errors.

Page 6

"N-CaM forms only a 1/1 complex forms… ": delete 2nd "forms".

page 6

"At trace ca^2+^, the W432 spectra obtained with SKp bound to C-CaM is indistinguishable from when SKp is bound to WT-CaM (Figure 4D) … The spectra with SKp bound to N-CaM appears slightly more like the spectra of SKp bound to WT-CaM, but not enough to conclude that the N-lobe of CaM binds W432 when full-length WT-CaM binds SKp. "

Difficult to make these comparisons in the figure. Do a comparison? Show a statistic.

Page 6 "10 μM ca^2+^ is sufficient to fully activate SK2 channels (Figure 5)." Not clear that Figure 5 demonstrates this.

Discussion

Page 13 "…a more complex model is necessary to truly represent the mechanism for SK gating by WT-CaM."

As the manuscript never shows that ca^2+^ binding to CaM C-lobe contributes SK conductance activation it seems this sentence should be softened. Suggestions:

"a more complex model is likely necessary to truly represent the mechanism for SK gating by WT-CaM" or "a more complex model is necessary to truly represent the mechanism for SKp binding by WT-CaM".

Figure 1, indicating precisely where the SKp (residues 396-487 of rat SK2) correspond in Panels A and B would be helpful.

Figure 3 legend: what are colored arrowheads?

Figure 4

While I can see the differences in the waveforms the authors refer to in the results, it might be difficult for others to do so. This figure could benefit from some form of direct comparison between conditions. Possibly plots or tables of peak wavelengths, and amplitudes?

Figure 6A, B

Unclear what ratios the values in between tick marks correspond to.

Figure 7B

The cartoon seems to indicate preformed complexes are not with WT-CaM.

*Reviewer #2 (Recommendations for the authors):*

1. The results are described in a somewhat nonlinear fashion which makes it challenging to understand the findings and their relevance to the overall model. For example, Figure 4D is described after parts of Figure 6. This makes it very difficult to follow the complex arguments. In part, this is because the figures are arranged according to the experimental technique. I wonder if it would be simpler to organize based on the specific question that is answered?

2. In Figure 5, the authors demonstrate that the addition of N-X at least partially activates the channel. Does the exogenous addition of X-C or X-X have any effect? These may be helpful to delineate any potential functional role for association of ca^2+^ bound C-lobe as one may envision based on in vitro analysis

3. The requirement of one N- and one C- lobe for activation of SK channels is very interesting. Will shuffling the arrangement of N and C also support channel activation, i.e could C-N CaM also activate the channel as effectively as N-C, or is the arrangement of the lobes also important?

4. The MALS assays are quite impressive in terms of their ability to resolve distinct CaM lobe-peptide complexes. That said, the prevalence of the C-P-C complex is quite low, I am curious if one were to assume that such complexes do not form, how much worse would the fits be for the simulated Molar Mass curves in Figure 3?

5. Figure 7B shows P-N with N or P-C with C, but the description in Pg. 7 states that "Using preformed P-C with WT-CaM, …" Is the figure mislabeled, or was the complex formed with N or C lobe?

6. Pg 2. "In includes ca^2+^ binding to CaM, …" should be "It includes"

7. Pg. 12. "One limitation of our study is that a peptide is free to bind CaM in configurations that are more constrained than it would be in a membrane-bound channel."

I am not sure about the intended meaning here particularly with "constrained,": On the one hand if portions of the peptide are not exposed in the full-length channel then I'd imagine CaM can interact in conformations that are not otherwise available making it less constrained. On the other hand CaM additional sites beyond what is encoded in the peptide become available for CaM interaction. Is the sentence suggesting both?

8. As noted above, a major shortcoming is that the overall mechanism of channel activation is not clear, in particular how the new studies revise previous models. For example, although C-P-C complex can form in solution based on MALS and on titration of CaM to CaM-P preformed complexes, the authors note that it is unlikely in the context of the full-length channel.

*Reviewer #3 (Recommendations for the authors):*

1. The authors rely heavily on a stoichiometry assay, drawing multiple conclusions from the existence (or lack thereof) of the P-C-P complex. In fact, the summary figure is a model of entry into the 2:1 and 1:2 binding complexes. Yet there is no discussion of the relevance of this complex. Are the authors asserting that there is a change in the stoichiometry of the CaM/channel complex in response to ca^2+^?

2. The overall organization of the text and figures is unnecessarily onerous. The figure is not presented in order and I found I was constantly flipping from figure to figure as I read through the text.

3. The patch clamp data is a strong feature of this paper, yet it is presented at such a condensed scale that it is difficult to read. In particular, in Figure 2 where the data is first presented, the data is smaller than the cartoons describing the experiment which actually makes it difficult to decipher the experimental protocol.

4. No statistical tests are performed in the paper. Statistical significance should be added to figure 5 and table 2.

5. Table 2 describes the affinity measured for the CG-MALS experiments, but the text focuses almost exclusively on the stoichiometry results (which may or may not be functionally relevant in a holochannel). Is there a reason that affinity is not a more relevant result?

6. P2, 3rd line of the last paragraph. The word 'calcium' is missing after 10 µM.

[Editors' note: further revisions were suggested prior to acceptance, as described below.]

Thank you for resubmitting your work entitled "Calcium dependence of both lobes of calmodulin is involved in SK channel gating" for further consideration by *eLife*. Your revised article has been evaluated by Kenton Swartz (Senior Editor) and a Reviewing Editor.

The manuscript has been improved but there are some remaining issues that need to be addressed, as outlined below:

The revised version of this beautiful study is clearer and even more appropriately nuanced. We appreciate how findings with SKp are carefully distinguished from gating in the body of the manuscript, and the title could would benefit from similarly precise wording.

The title "Calcium dependence of both lobes of calmodulin is involved in SK channel gating" seems oversimplified, as the data showing calcium dependence involved the SKp peptide, with no channel to gate. We ask that you to revise the title. One suggestion would be: "Calcium dependence of both lobes of calmodulin is involved in interactions with a domain of SK channels".

---

## [Author Response]

Essential revisions:1) Please make it abundantly clear to readers (e.g., in the Abstract and Discussion) that while the results suggested that ca^2+^ binding to the C-lobe causes a conformational change, electrophysiological experiments did not conclusively show that ca^2+^ binding to the C-lobe of CaM impacts SK channel gating. In the Abstract and end of the Discussion, the wording seems to project conclusions from soluble SKp onto SK conductance activation without a disclaimer. It might help to expound on the limitations of projecting results with SKp onto gating of intact SK channels.

The referees for the manuscript were both professional and insightful. We appreciate their thoughtful reviews and for our chance to respond to them.

Peptide work lays the foundation for interpreting full-length protein data. Full length proteins have more binding sites, and therefore more parameters to be determined making it extraordinarily difficult to solve mechanistic problems without breaking the problems down into solvable parts. While it is true that binding energies may be different in a whole protein compared to its parts, it may be possible with multiple studies to figure out how smaller parts relate to the whole to make a universally consistent mechanistic model. Our work by itself cannot complete a full mechanistic model any more than any other single study. We can assert that the prevailing gating models of a full-length channel protein are at best incomplete.

The reviewer comments were helpful, and we clarified our position the abstract, introduction and in the discussion.

We reworded the abstract to clarify that we arrived at our conclusions from work using peptide binding studies, but electrophysiology on full-length channel.

2) The conclusion that the X-C CaM – SKp complex changes conformation in ca^2+^ is important, but difficult to see in Figure 4. Please, provide a quantitative comparison of the data in Figure 4 which leads to important conclusions, so that readers who may struggle to overlay emission spectra in their heads can benefit.

We added comparisons in the results to contrast spectral peak location and intensity to assist the reader. We also added measured peak positions to figure 4. We hope this will aid the reader following our report. Our report did need to be clearer. Thank-you for suggesting that we clarify our findings.

3) Please address each reviewer's concern with changes to the manuscript, or explain why not in the rebuttal.

We did as we were instructed. We appreciate all comments from each reviewer.

Reviewer #1 (Recommendations for the authors):Despite trying hard to find flaws in the logic of interpretation, I've come up empty-handed. The interpretations seem logically watertight, with the exception of a few wording choices in the abstract and final paragraph.My biggest suggestion is to make the identity of SKp and the limitations of using only the C-terminal peptide more obvious in Figure 1 and the introduction.One interesting finding that might be worth discussing further is that the change log KA was similar for WT-CaM and N-CaM (Table 2). Maybe this is expected, but could this be interpreted as evidence that the energy of ca^2+^-induced binding of N-CaM is sufficient to account for the binding of WT-CaM?

This brings up a good point that we needed to make clearer. In the presence of ligand, ca^2+^ in our case, protein binding can be complicated. We emphasize that CaM is a ca^2+^ sensor with four binding sites. We added in the manuscript that additional ca^2+^-binding studies are needed to draw comparisons between the Ka of mutant or WT-CaM with SKp. We also added that WT-CaM has additional ways to bind SKp at high ca^2+^, again implying that the C-lobe is involved.

D->A mutations in the C-lobe 3,4 sites decrease ca^2+^ sensitivity and Hill coefficient of SK activation (Xia et al., 1998). It could be helpful to discuss whether this prior finding provides evidence that the C-lobe ca^2+^ sites are important for binding or efficacy.Specific comments:Abstract"widely accepted": suggest removing social speculation.

Line 7 changed wording to “previous”

"both lobes bound to SK are ca^2+^ sensitive": suggest changing "SK" to "SKp".

The sentence that contained this phrase was changed for the “Essential Revisions” section above. We note that we intentionally used SK2, but we preface the use of SK2 with “We interpret…”.

IntroductionPage 1"…which results…" → which can result?

We changed the phrase to, “which can result”.

ResultsPage 5"Intrinsic tryptophan fluorescence of SKp confirms calcium sensitivity in both lobes of calmodulin."The results showing this are not discussed in this section. Only WT CaM is discussed here.

Indeed, our subtitle was unchanged from an early version and we missed updating it when the section was split. We removed “in both lobes” from the subtitle. Thank-you for catching this oversight.

Page 5…peak is shifted towards shorter wavelengths (329 nm)…please provide a quantitative estimate of shifts and errors.

We provided more values to help the reader make comparisons between spectra. This was a very important comment, and we feel there is work one could do to improve on this technique in the future.

The combination of the instrument noise on top of the protein noise made it difficult to produce believable error bars for position using standard approaches. We decided to only compare the averaged spectra, but in doing so, we lose the ability to quantify the standard error of the spectral peak position. Our discussion of spectra is more qualitative, but it should be obvious when the spectra are fundamentally different.

Error analysis of peak location proved difficult with our data sets. The raw spectra had two kinds of noise that were difficult to treat equally across the different spectral appearances. Additionally, a decreased intensity at ‘bluer’ wavelengths indicated that the 310 nm cut-on filter distorted some of our results.

The greatest noise source was from protein across different batches of SKp. The smaller errors in panels B, E, and H were because the SKp were from the same batches. The concentration of SKp is always determined the same way using uv absorbance spectra; however, in the fluorimeter, nearly identical concentrations by uv absorbance did not produce identical tryptophan emission spectra intensity, although the behaviors in complexes are generally the same across experiments. Regardless, we provide our error ranges enabling us to more firmly generalize our findings.

The second noise source was from the instrument. Signal averaging the spectra over longer times would have certainly helped with error analysis, but we have strong enough reproducible data to show that our general conclusions are valid.

Cut-on filters are essential to block scattered excitation light to the detector. Cuton filters are not perfect and a 310 nm cut-on filter blocks fluorescent light nonlinearly from 310 to ~340. This blocking would increasingly distort intensity closest to 310 nm, so a blue shifted curve is affected more. If filtering distortions could be corrected, it is likely that greater spectral differences between EGTA and ca^2+^ would be unmasked.

Page 6"N-CaM forms only a 1/1 complex forms… ": delete 2nd "forms".

We corrected this Max Headroom error.

page 6"At trace ca^2+^, the W432 spectra obtained with SKp bound to C-CaM is indistinguishable from when SKp is bound to WT-CaM (Figure 4D) … The spectra with SKp bound to N-CaM appears slightly more like the spectra of SKp bound to WT-CaM, but not enough to conclude that the N-lobe of CaM binds W432 when full-length WT-CaM binds SKp. "Difficult to make these comparisons in the figure. Do a comparison? Show a statistic.

As discussed above, we added more comparisons.

Page 6 "10 μM ca^2+^ is sufficient to fully activate SK2 channels (Figure 5)." Not clear that Figure 5 demonstrates this.

The reviewer is correct; we removed the figure call and inserted a reference that supports our statement. The reference has several calcium dose-response relations showing that WT-SK activity saturates at 10 μM ca^2+^.

DiscussionPage 13 "…a more complex model is necessary to truly represent the mechanism for SK gating by WT-CaM."As the manuscript never shows that ca^2+^ binding to CaM C-lobe contributes SK conductance activation it seems this sentence should be softened. Suggestions:"a more complex model is likely necessary to truly represent the mechanism for SK gating by WT-CaM" or "a more complex model is necessary to truly represent the mechanism for SKp binding by WT-CaM".

Agreed, we added ‘likely’ to the sentence.

Figure 1, indicating precisely where the SKp (residues 396-487 of rat SK2) correspond in Panels A and B would be helpful.

Added the sentence “The peptide SKp includes HA, HB, and part of HC.” to the figure legend.

Figure 3 legend: what are colored arrowheads?

Arrows were defined.

Figure 4While I can see the differences in the waveforms the authors refer to in the results, it might be difficult for others to do so. This figure could benefit from some form of direct comparison between conditions. Possibly plots or tables of peak wavelengths, and amplitudes?

Direct comparisons were included. Good suggestion.

Figure 6A, BUnclear what ratios the values in between tick marks correspond to.

This may be a struggle with display. There were more in-between data points with these sets. There was not enough space to include tick marks that clearly indicate molar ratio at each data point, but hopefully they can be estimated by the reader.

Figure 7BThe cartoon seems to indicate preformed complexes are not with WT-CaM.

We are relieved this was brought to our attention. Thank-you.

Reviewer #2 (Recommendations for the authors):1. The results are described in a somewhat nonlinear fashion which makes it challenging to understand the findings and their relevance to the overall model. For example, Figure 4D is described after parts of Figure 6. This makes it very difficult to follow the complex arguments. In part, this is because the figures are arranged according to the experimental technique. I wonder if it would be simpler to organize based on the specific question that is answered?

During writing, we struggled with how to best organize the topics. In the end, we felt it was most important to keep certain figures together rather than fragment them. For example, the panels in figure 4 are arranged so that all can be compared with each other, even though the mutants are discussed later. The comparisons would be less obvious to see if they were separate figures. The paper is organized by theme, which we felt was most appropriate. We apologize if this is still too cumbersome, there was a lot of data that had to be distilled.

2. In Figure 5, the authors demonstrate that the addition of N-X at least partially activates the channel. Does the exogenous addition of X-C or X-X have any effect? These may be helpful to delineate any potential functional role for association of ca^2+^ bound C-lobe as one may envision based on in vitro analysis

Good question! We published in Li et al. most of our answer, but there are still unsolved parts. Exogenous X-C was able to rescue current with additional SK agonist EBIO or NS-309, or with Terbium, which is a lanthanide ion that binds stronger to the EF-hands of CaM than ca^2+^. Unfortunately, there are still some unanswered binding questions, so we did not go into detail X-C in this report. This is an exciting avenue to pursue!

3. The requirement of one N- and one C- lobe for activation of SK channels is very interesting. Will shuffling the arrangement of N and C also support channel activation, i.e could C-N CaM also activate the channel as effectively as N-C, or is the arrangement of the lobes also important?

We really wanted to know the answer to this question too. We don’t have data.

4. The MALS assays are quite impressive in terms of their ability to resolve distinct CaM lobe-peptide complexes. That said, the prevalence of the C-P-C complex is quite low, I am curious if one were to assume that such complexes do not form, how much worse would the fits be for the simulated Molar Mass curves in Figure 3?

The C-P-C is the complex with the largest molar mass that we observe, but as you noticed it is not very abundant. MALS is extremely biased towards detecting larger complexes compared with small ones. Thus, in panel A, simulated molar mass cannot produce the ‘M’ shape observed in the raw data without the C-P-C being one of the complexes that can form. The structure of the ‘M’ gives us absolute certainty that C-P-C is present in panel A.

We have to be more guarded in panels B and C. I hope we expressed that there is probably C-P-C and not certainly C-P-C in those cases. We added a little more to clarify this and to hopefully avoid confusion.

5. Figure 7B shows P-N with N or P-C with C, but the description in Pg. 7 states that "Using preformed P-C with WT-CaM, …" Is the figure mislabeled, or was the complex formed with N or C lobe?

We are relieved this was brought to our attention. Thank-you. The figure was mislabeled.

6. Pg 2. "In includes ca^2+^ binding to CaM, …" should be "It includes"

We corrected this, thank-you.

7. Pg. 12. "One limitation of our study is that a peptide is free to bind CaM in configurations that are more constrained than it would be in a membrane-bound channel."I am not sure about the intended meaning here particularly with "constrained,": On the one hand if portions of the peptide are not exposed in the full-length channel then I'd imagine CaM can interact in conformations that are not otherwise available making it less constrained. On the other hand CaM additional sites beyond what is encoded in the peptide become available for CaM interaction. Is the sentence suggesting both?

You are correct that we meant “less” constrained. We fixed that wording error.

8. As noted above, a major shortcoming is that the overall mechanism of channel activation is not clear, in particular how the new studies revise previous models. For example, although C-P-C complex can form in solution based on MALS and on titration of CaM to CaM-P preformed complexes, the authors note that it is unlikely in the context of the full-length channel.

We understand that we cannot elucidate a full-channel mechanism, even with all our electrophysiology and binding data from this work and prior. However, we claim the same argument applies equally to all studies without putting their work into context of the decades of research on this topic. We are cautious because we perceive gaps in our knowledge. More research in all fields is required to develop even the simplest model for how these channels work, especially in the closed state dwell times. We attempted to explain this in our first draft, but from these reviewer’s helpful comments we have attempted to attract more debate, and thus research, on calmodulin and ion channels.

Reviewer #3 (Recommendations for the authors):1. The authors rely heavily on a stoichiometry assay, drawing multiple conclusions from the existence (or lack thereof) of the P-C-P complex. In fact, the summary figure is a model of entry into the 2:1 and 1:2 binding complexes. Yet there is no discussion of the relevance of this complex. Are the authors asserting that there is a change in the stoichiometry of the CaM/channel complex in response to ca^2+^?

We hope others in the field will ask this same crucial question. The key matter is that ca^2+^ alters the binding of CaM to SKp, and this finding most likely applies to full-SK as well. The “rescue” experiments show the requirement for ‘full-length' CaM for SK function, but the peptide-binding experiments show that mutant CaMs have altered interactions with target proteins.

2. The overall organization of the text and figures is unnecessarily onerous. The figure is not presented in order and I found I was constantly flipping from figure to figure as I read through the text.

We apologize for that the data were not perceived as the optimal order. There is simply a lot of data that needs to compared within single figures. That required us to refer to earlier figures in the paper. We prefer the order of logic starting with WT protein and proceeding through mutants at a later stage.

3. The patch clamp data is a strong feature of this paper, yet it is presented at such a condensed scale that it is difficult to read. In particular, in Figure 2 where the data is first presented, the data is smaller than the cartoons describing the experiment which actually makes it difficult to decipher the experimental protocol.

We corrected the scales of the panels to emphasize the date more than the cartoon. We agree that this has a better appearance. We that the reviewer for this suggestion.

4. No statistical tests are performed in the paper. Statistical significance should be added to figure 5 and table 2.

We added t-test values for Figure 5. Also in the text, p-values were added where comparisons are made.

5. Table 2 describes the affinity measured for the CG-MALS experiments, but the text focuses almost exclusively on the stoichiometry results (which may or may not be functionally relevant in a holochannel). Is there a reason that affinity is not a more relevant result?

The CG-MALS data are robust for stoichiometry and for guiding future binding assays. For certain stoichiometries, the measured affinities are simply bound by the limits of the instrument as we stated. We expect that future binding assays will validate our measurements, but we are not fully confident that all the values are accurate without deeper testing (hence the ‘italicized’ values in Table 2). The reviewer is correct that accurate and precise measures of affinity are needed, but it is beyond the scope of this work.

6. P2, 3rd line of the last paragraph. The word 'calcium' is missing after 10 µM.

We corrected this, thank-you.

[Editors' note: further revisions were suggested prior to acceptance, as described below.]

The manuscript has been improved but there are some remaining issues that need to be addressed, as outlined below:The revised version of this beautiful study is clearer and even more appropriately nuanced. We appreciate how findings with SKp are carefully distinguished from gating in the body of the manuscript, and the title could would benefit from similarly precise wording.The title "Calcium dependence of both lobes of calmodulin is involved in SK channel gating" seems oversimplified, as the data showing calcium dependence involved the SKp peptide, with no channel to gate. We ask that you to revise the title. One suggestion would be: "Calcium dependence of both lobes of calmodulin is involved in interactions with a domain of SK channels".

The request by the editors to make our title more precise is reasonable and we have changed our title. Our title now reads “Calcium dependence of both lobes of calmodulin is involved in binding to a cytoplasmic domain of SK channels.” We hope this addresses the concern.